# Improvement of One-Shot-Learning by Integrating a Convolutional Neural Network and an Image Descriptor into a Siamese Neural Network

Jaime Duque Domingo *, Roberto Medina Aparicio and Luis Miguel González Rodrigo

CARTIF Foundation, División de Sistemas Industriales y Digitales, Parque Tecnológico de Boecillo, 47151 Valladolid, Spain; robmed@cartif.es (R.M.A.); luirod@cartif.es (L.M.G.R.)
* Correspondence: jaiduq@cartif.es

**Abstract:** Over the last few years, several techniques have been developed with the aim of implementing one-shot learning, a concept that allows classifying images with only a single image per training category. Conceptually, these methods seek to reproduce certain behavior that humans have. People are able to recognize a person they have only seen once, but they are probably not able to do the same with certain animals, such as a monkey. This is because our brains have been trained for years with images of people but not so much of animals. Among the one-shot learning techniques, some of them have used data generation, such as Generative Adversarial Networks (GAN). Other techniques have been based on the matching of descriptors traditionally used for object detection. Finally, one of the most prominent techniques involves using Siamese neural networks. Siamese networks are usually implemented with two convolutional nets that share their weights. They receive two images as input and can detect whether they belong to the same category or not. In the field of grocery products, there has been a lot of research on the one-shot learning problem but not so much on the use of Siamese networks. In this paper, several classifiers are firstly evaluated to decide on a convolutional model to be used with the Siamese and to improve the baseline results obtained in the dataset used. Then, two existing techniques are integrated within the Siamese model: a convolutional net and a Local Maximal Occurrence (LOMO) descriptor. The latter was initially used for the re-identification of people although it has shown its effectiveness to improve the values of a traditional Siamese with only convolutional sisters. The whole network is trained on categories and responds to different categories, showing its strong capacity to deal with the problem of having only one image per category.

**Keywords:** one-shot learning; siamese; ResNeXt-101; LOMO; grocery image classification

## 1. Introduction

Computer vision is one of the disciplines that have seen the greatest increase in the number of applications in recent years. The development of deep learning along with the increase in computing capabilities has made it possible to solve problems that were previously difficult using traditional techniques. A key aspect for the good performance of modern image classifiers is the number of training samples. While a human being is able to recognize an object after only seeing it once, computers require datasets with a lot of images per class to be trained and deliver acceptable results. On an industrial level, image processing that requires a single example to be implemented is a desire and a need that, until now, could rarely be fulfilled. These systems expand the possibilities for implementing new visual inspection systems in the industry.

Siamese networks are one of the techniques that can deal with the problem of *one-shot learning* [1], where there is only one image per category. They are commonly implemented with two convolutional networks that share their weights. They receive two images as input and can detect whether they belong to the same category or not. Although different

methods have been applied to the grocery products classification, there has not been so much research on the use of Siamese nets. In this paper, two existing techniques are integrated within the Siamese model: a convolutional net and a Local Maximal Occurrence (LOMO) descriptor. Liao et al. [2] presented the LOMO descriptor as a method for the Re-ID problem (people re-identification), but our paper has shown its effectiveness for improving the values of a traditional Siamese with only a convolutional sister. Several classifiers have been first evaluated to decide on a convolutional model to be used with the Siamese. This model also improves the baseline results obtained in the baseline of Grocery Store Dataset [3]. The whole network is trained on categories and responds to new different classes, showing its strong capacity to deal with the problem of having only one image per category.

The present paper is structured as follows: Section 2 explores the state-of-art technologies considered in this paper. Section 3 describes the two procedures carried out. The first procedure evaluates different classification models that are valid for the Siamese net and improves the results stated in Grocery Store Dataset [3]. The second procedure presents the Siamese net integrating a ResNeXt-101 [4] and a LOMO descriptor. Different regularization mechanisms are evaluated. In Section 4, the different experiments and results obtained with the system are reported. An overall discussion on the obtained results is set out. Finally, Section 5 notes the advantages and limitations of the presented system and suggests future developments.

## 2. Overview of Related Work

*Few-shot Learning* (FSL) [5] is the name given to a group of techniques that, using prior knowledge, can rapidly generalize to new tasks containing only a few samples with supervised information. Among them, image classification techniques for learning from a single example are named *one-shot learning* (OSL) [1]. These techniques make it possible to train models with many different classes and only one image per class. Common classification models require a large battery of images for each category whereas OSL looks for models with a single image per class, in an ideal model or a small number of images. One-shot learning techniques are based on the idea of inducing new knowledge from previously obtained knowledge by using a classifier trained with similar cases, emulating to some extent the way the human brain learns new ideas using knowledge from previous experiences. As an example, a human distinguishes the face of a person only seen once because we have prior information about many faces. However, it is much more difficult for us to distinguish two individuals of other species without having any prior knowledge. An outstanding work proposed by Held et al. [6] explored the idea of training a classifier with a different group of categories for train-test. The authors wanted to recognize a grocery product for which only a single image had been given, but they also wanted to consider novel viewpoints. They performed a multi-stage training procedure in which they first trained on a large class-level dataset, followed by an auxiliary multi-view dataset, which allowed the model to be robust relative to viewpoint changes. Finally, they trained on the objects they wanted to recognize from just a single image.

### 2.1. Solving the One-Shot Learning Problem

There are different strategies that have been developed to solve the problem of OSL. The first strategy, called *data augmentation* [7], consists of increasing the number of images in the dataset by carrying out simple transformations on them. For example, this method allows obtaining large number of images for a category with only a single image of that class by performing transformations on it (translations, rotations, changes in illumination, deformations, etc.). This technique is integrated with many data generators that feed images to the training methods.

A second technique consists of generating fictitious databases from knowledge extracted from classes of similar objects. Within this group, *Generative Adversarial Networks* (GAN) [8–10] are capable of generating images of unknown classes, such as the case of a

face of an unknown person. Regarding the grocery products, Tonoono and Di Stefano [11] proposed an end-to-end architecture comprising a GAN with a generator and a discriminator. The GAN augmented the training set with samples similar to those belonging to the test domain while simultaneously produced hard examples for the embedding network. They performed recognition by means of a k-Nearest Neighbor (kNN) search against a database consisting of only one reference image per product. The same approach using data augmentation generation by means of a GAN has been recently proposed by Wei et al. [12].

A third technique uses a probabilistic approach and consists of calculating the probability that an object belongs to a specific class by analyzing the characteristics of the image that have been useful in the classification of objects of the same type [13]. This probability is obtained by calculating the distance between the feature vector of a new object and those in the available database. In order to introduce a new class into the system for which only a single image is available, the feature vector of the new class must meet certain characteristics in order to be distinguishable from the rest.

A fourth technique consists on feature extraction and matching, creating descriptors such as SIFT [14], SURF [15] or ORB [16]. These methods are slow because matching has to be performed with all possible images. In addition, they do not generalize as well as other methods. Regarding this technique, a fine-grained grocery product recognition method has been explored by Geng et al. [17], who addressed the OSL problem by presenting a hybrid classification approach that combined feature-based matching and one-shot deep learning with a coarse-to-fine strategy. The candidate regions of the product instances were firstly detected and coarsely labeled by recurring features in the product images without any training. They generated attention maps to guide the classifier to magnify the influences of the features in the candidate regions. However, the authors used a dataset with grocery packages that did not include products without fixed landmarks, such as fruits. They selected the logo regions on product packages to be able to clearly match descriptors.

Finally, a fifth technique involves the use of neural networks and, more specifically, *Siamese Neural Networks* (SNNs) [18]. SNNs usually compare the features output of two convolutional nets to infer whether two images belong to the same class or not. The comparison is carried out using the feature vectors obtained before the last classification layers. Each of the two input images is channeled through one of the convolutional nets, which share models and weights. They started to be used by Bromley et al. [19] in signature verification work. They have also been used for different kinds of problems, such as the evaluation of source code similarity [20], cyber attack detection [21], object tracking [22], chromosome classification [23] and even animal sound classification [24]. More recently and regarding the grocery products, Ciocca et al. [25] have applied a Siamese network to capture the relations between iconic and natural images in the Grocery Store Dataset [3]. They evaluated several Siamese models with different Convolutional Neural Networks (CNNs), obtaining the best results with a DenseNet-169 [26] backbone.

### 2.2. Integration of Multiple Classifiers

The integration of multiple classifiers or object detectors has been previously explored. Xue et al. [27] developed a sort of Siamese net that integrated features from two nets: a VGG-16 to perform a contour detection and a YOLOv3 to detect objects. This network was able to track objects benefiting from both classifiers. However, this model was not proposed for an OSL problem.

### 2.3. Grocery Datasets

Regarding the grocery store products, several datasets have been created during the last few years, such as the Grocery Store Dataset [3], the MVTec D2S dataset [28], the Retail Product Checkout dataset (RPC) [29] or the Freiburg groceries dataset [30]. Among them, MVTec D2S, RPC and Freiburg datasets are focused on the problem of object detection rather than classification. The Grocery Store Dataset [3] contains image data of grocery items in fine and coarse categories and is valid for our OSL and classification problems.

The Grocery Store Dataset is composed of 5125 images of 81 different kinds of fruit, vegetables and carton items (e.g., juice, milk and yogurt). All images were taken with a smartphone camera in different grocery shops. In addition to the fine categories (81), there are 43 coarse categories where, for example, the fine class *Royal Gala* and *Granny Smith* belong to the same coarse class *Apple*. In addition to the natural images, there are iconic images, which represent the product taken in controlled lighting conditions and without the supermarket background. In the classification experiments, the authors have connected the feature vector prior to the connection of the dense layers to an SVM classifier. By testing different CNNs such as AlexNet, VGG16 or DenseNet, the authors obtained 72.5% accuracy, using directly a model trained with ImageNet [31] and 85% accuracy on test using a model on which they performed fine tuning. In these winner cases, the authors used a DenseNet-169. In the same way, they tested a DenseNet-169 classification network without using SVM. In this case, they obtained 84% accuracy. Our first experiments have consisted in improving their classification baseline to drive the creation of our Siamese. Next, different configurations of the combined Siamese net are evaluated to show the improvement over using a traditional Siamese with only CNNs.

## 3. Analysis of the System

Two procedures have been carried out. The first procedure explained in Section 3.1 consisted of searching for a CNN able to improve the results stated in the classification baseline of the Grocery Store Dataset [3] paper. This result helped us designing the Siamese network. The second procedure explained in Section 3.2 describes the integration of a classifier and a descriptor into the Siamese model, improving the results obtained in the OSL approach with only a CNN.

### 3.1. Improving the Classification Base Line

Neural networks for image classification allow inferring which category an input image belongs to. These networks use layers of different types, including convolutions that apply a given convolution matrix (kernel) with a sliding window over the image or pooling layers that allow the dimensionality of matrices to be reduced, e.g., by obtaining the maximum of a sliding $2 \times 2$ matrix. Deep neural networks use many consecutive convolution layers to capture salient elements of the images, from the most generic to the most particular.

One of the early problems with deep networks was that the accuracy started to saturate at one point and eventually degrade. In addition, the model did not converge due to vanishing gradients. These problems were partly solved by the use of residual blocks [32] that connect the input of a block to the output of that block via an aggregation.

Another problem with deep convolutional networks was the high growth of the number of parameters when the number of layers increased. The ResNet architecture [33] included residual bottleneck blocks. This model was a variant of the residual block, which uses $1 \times 1$ convolutions to create a bottleneck. The use of a bottleneck reduces the number of parameters and matrix multiplications without noticeably altering the result. The idea was to make the residual blocks as thin as possible to increase the depth and to possess fewer parameters.

The ResNeXt-101 [4] is based on a ResNet model but replaces the $3 \times 3$ convolutions within the ResNet model by $3 \times 3$ convolutions grouped together. This grouping is a technique inherited from AlexNet [34]. The ResNeXt-101 bottleneck block splits a single convolution into multiple smaller parallel convolutions. The concept of cardinality refers to the number of parallel convolutions. As an example, in ResNeXt-101-32x8d, the cardinality is equal to 32 (number of parallel convolutions), and the bottleneck width is equal to eight (number of convolution filters). A notable difference from ResNet models is that ResNeXt uses aggregation instead of concatenation in the original Inception-ResNet block.

In order to improve the results given by Klasson et al. [3], two recent convolutional networks have been evaluated. ResNet-152 [33] and ResNeXt-101 [4] have been trained

to classify the 81 supermarket products. As shown in the Experiments Section 4, these networks directly improve on the baseline set by the authors. The networks have been modified to add a dropout before *Full-Connected* layers in order to increase generalization and improve the results.

During the experiments, we noticed that two fruits, oranges and satsumas became very confused as they were very similar. In order to overcome this problem, a cascade classifier has been proposed, as shown in Figure 1, to classify oranges and satsumas again if the result of the first classifier was one of these two fruits. The result of the first classifier has 81 outputs $(c_1, c_2, \ldots, c_{81})$, while the second classifier only has two outputs $(c_1{}'$ and $c_2{}')$.

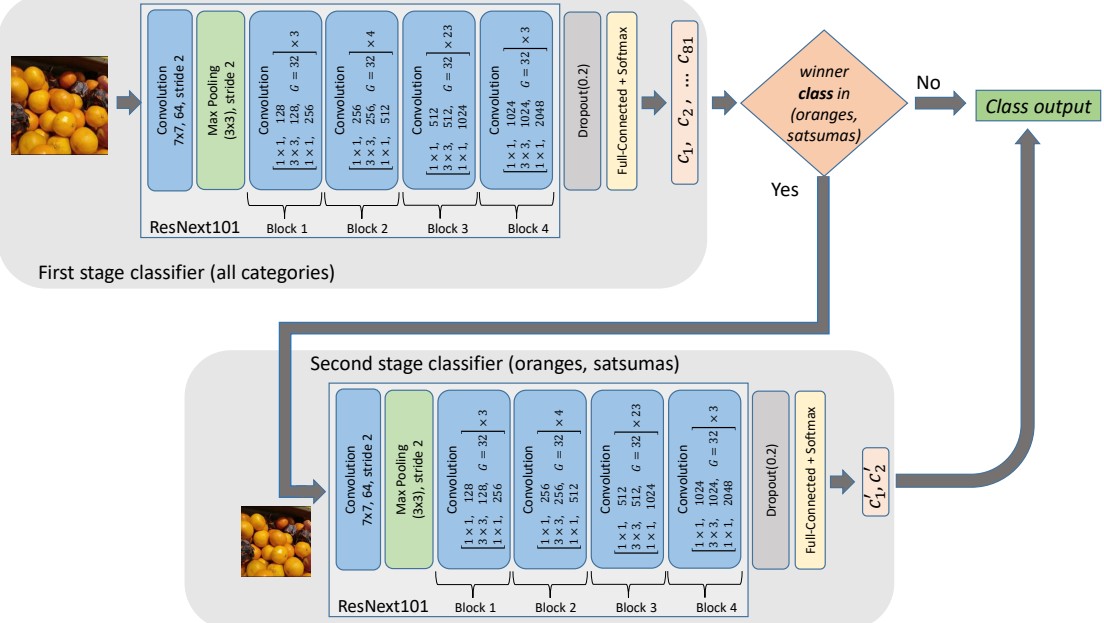

**Figure 1.** Cascade classifier based on ResNeXt-101-32x8d.

The classification results obtained during this first part allowed us to choose a model to continue with the OSL problem using Siamese nets.

### 3.2. Integration of Multiple Classifiers into a Siamese Network

The second proposed procedure allows using Siamese networks to solve the one-shot learning problem. In an OSL problem, a single image is available for each category. An approach to solving this problem consists of training the model with some classes of the dataset and to validate it with a different group of classes. The test is finally carried out with a third group of classes. Figure 2 shows the category split that has been made. There are 50 classes for training, 18 for validation and 13 for test (81 totals).

Three approaches based on Siamese nets have been evaluated: a traditional Siamese net with a CNN, a Siamese net using a LOMO descriptor [2] and a Siamese net integrating a CNN and a LOMO descriptor.

In the first approach to solving the problem, a Siamese net with a ResNeXt-101-32x8d [4] has been used. A Siamese net compares the feature layer of two CNNs to infer whether two images belong to the same class or not. Each of the two input images of the network is channeled through one of the convolutional sisters, which share models and weights.

Figure 3 shows the scheme of this architecture. Each of the sister subnets shares the same ResNeXt-101 model with the same parameters. The model receives two images as input. The convolutional part of the ResNeXt-101 produces a feature vector in the last layer $(u_1, u_2, \ldots, u_{64})$. In a classification problem, this vector is connected to the Full-Connected (FC) layer with a *Softmax* activation. However, in this case, this vector is

transformed into a single dimension *(Flatten)* and compared by Euclidean distance with the vector produced for another image $(v_1, v_2, \ldots, v_{64})$. In Figure 3, $d_E(u, v)$ represents the Euclidean distance. The distance is then connected to another layer with a single neuron and sigmoidal activation, responsible for inferring whether the two images belong to the same category "1" or whether they belong to a different category "0".

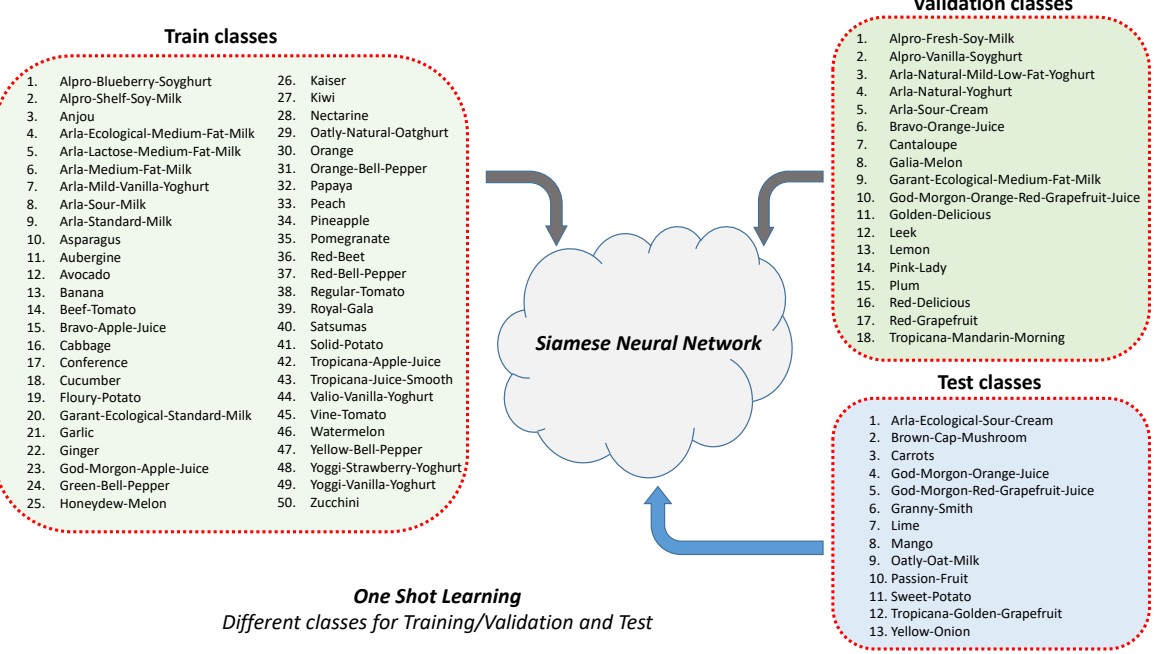

**Figure 2.** Category split to test the OSL problem.

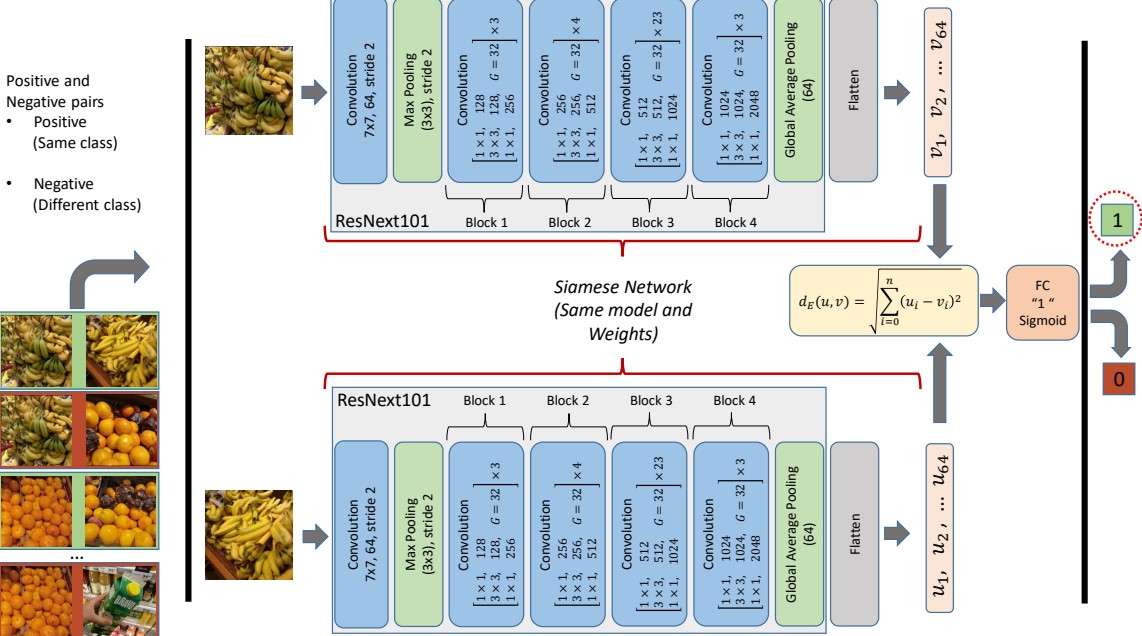

**Figure 3.** First approach: Siamese net with ResNeXt-101-32x8d.

In the second approach, a descriptor integrated within the Siamese net has been used (see Figure 4). In this case, the descriptor has no trainable parameters, but the rest of the model does. The LOMO descriptor was initially defined by Liao et al. [2] for the ReID problem (Re-identification of people). This descriptor first processes the images

with Retinex [35], with the aim of adjusting both the color constancy and dynamic range compression automatically, achieving a good approximation to human visual perception. From the images obtained with Retinex images, LOMO uses the HSV color histogram to extract color features. In addition, the Scale Invariant Local Ternary Pattern (SILTP) descriptor [36] is used for the description of the lighting invariant texture. SILTP is an improved operator of the well-known Local Binary Pattern (LBP) [37]. While LBP is invariant relative to certain transformations, it is not robust relative to noise. SILTP improves LBP by introducing a scale-invariant local comparison tolerance, achieving invariance relative to intensity scale changes and robustness relative to image noise.

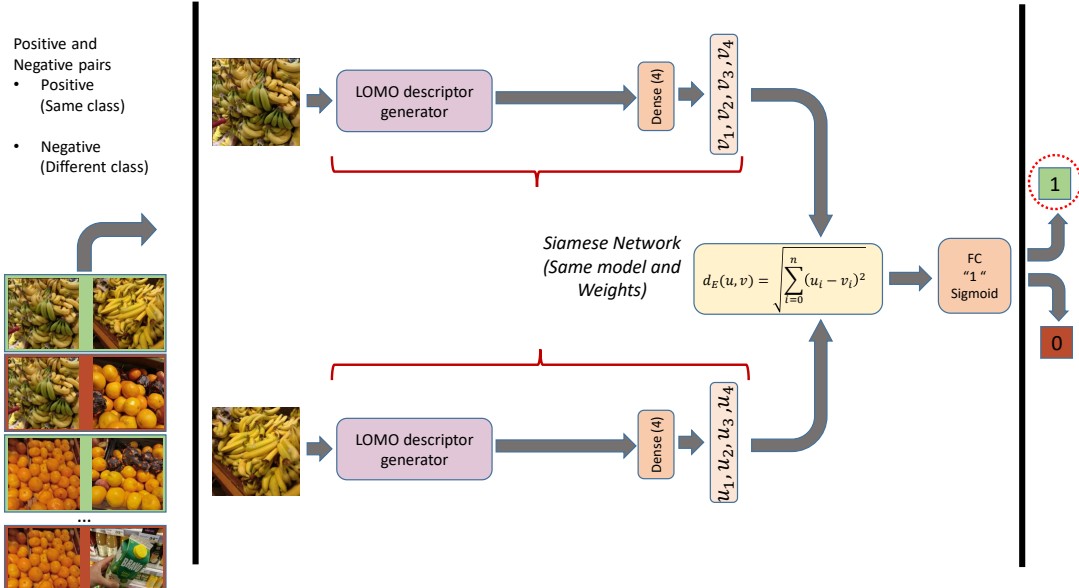

**Figure 4.** Second approach: Siamese net with LOMO descriptor.

LOMO works with $10 \times 10$ sliding windows, with an overlap of five pixels, to process local blocks in $128 \times 48$ images. Within each window, two SILTP histogram scales ($SILTP_{4,3}^{0.3}$ and $SILTP_{4,5}^{0.3}$) and a joint HSV histogram of $8 \times 8 \times 8$ cells are extracted. $SILTP_{N,R}^{\tau}(x_c, y_c)$ is equal to the concatenation of $N$ binary strings, obtained for $N$ neighborhood pixels equally spaced on a circle of radius $R$ with respect to the pixel $(x_c, y_c)$. The binary strings are calculated by using the $\tau$ scale factor. From $k = 0$ to $k = N - 1$, the binary string is "01" when $I_k > I_c \cdot (1 + \tau)$, "10" when $I_k < I_c \cdot (1 - \tau)$ and "00" otherwise. $I_k$ and $I_c$ are the gray intensities of pixels $k$ and $c$, respectively. Figure 5 graphically shows an example of how SILTP is calculated.

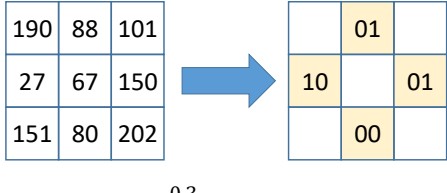

$$SILTP_{4,1}^{0.3} = 01100100$$

**Figure 5.** Example of SILTP code calculation.

Each cell of the histogram represents the probability of occurrence of a pattern in a sub-window. LOMO is able to deal with viewpoint changes because all sub-windows are checked at the same horizontal location, and the local occurrence of each pattern

is maximized. The resulting histogram manages to be invariant to viewpoint changes while, at the same time, captures the local characteristics of the bins of an image. In addition, the descriptor constructs a pyramid representation of three scales. Finally, the descriptor applies a logarithmic transformation to suppress the large values of the bins, and a normalization of the HSV and SILTP features is performed. As will be shown in the Experiments Section 4, this descriptor also produces results almost comparable to the CNN model used for our particular problem. Other descriptors that were evaluated, such as the known Histogram of Oriented Gradients (HOG), did not offer acceptable results.

A *dense* layer of four neurons was connected to the output of the descriptor (78,858 elements) and was, in turn, used to calculate the Euclidean distance. Although experiments were performed with different numbers of dense layers (two or three) and different numbers of neurons in those layers (2000, 1000, 500, 64, 16, 8, 4 and 2), the model performed slightly better than with other configurations with the value of four neurons. The use of a dense layer reduces the cost of directly comparing two vectors of 78,858 elements. It also improves the accuracy of the model. The choice of the hidden neurons and the number of hidden layers was made by a batch grid search, testing different configurations and keeping the best one. A value of hidden neurons was initially calculated following some *rule of thumb*, such as 2/3 size of input and output layer [38], or some used in previous works [39] that also related to the number of samples. However, taking as input 78,858 elements of the LOMO vector and as output four as the number of elements to be compared, the hidden neurons represented a value too high that caused our model to fail to load into the memory of our GPUs. Then, this number was lowered to 2000 neurons and hierarchically distributed into the different layers. Despite this *rule of thumb*, we found that the model worked best with a single layer and with only four neurons. This approach was also used in the next presented model.

Finally, in a third approach, a Siamese has been constructed by integrating a ResNeXt-101 and a LOMO descriptor on each side (see Figure 6). The idea of this model is that the network learns the strengths of each of the two methods. In this particular case, the output of the ResNeXt-101 and LOMO generator has been concatenated. Then, a dropout layer is applied to reduce overfitting with respect to the training set. Next, a *dense* layer with eight neurons is connected. This layer represents the new descriptor vector that is compared between two images. In fact, the sister net can operate in isolation to produce this vector, and it can be compared with other vectors to perform the classification. This model has been evaluated with and without regularization. Together with dropout, regularization is one of the mechanisms that reduce overfitting, bringing the training curve closer to the validation curve. Regularizers apply penalties on layer parameters during optimization. These penalties are summed into the loss function that the network optimizes. L2-regularization, also named *ridge regression*, adds the sum of the squared coefficients multiplied by a regularization factor to the loss function.

As in the previous model, different network configurations were evaluated. For example, in another case, the two descriptors of the LOMO images were concatenated, and their distances were calculated. On the other hand, they were connected to a layer of a single neuron. In the same manner, the distance of the ResNeXt-101 feature vectors was calculated, and the same vectors were concatenated and connected to a layer of one neuron. These single-neuron layers could discriminate the weight of each distance to consider either LOMO or ResNeXt-101. However, the selected architecture produced the most promising results.

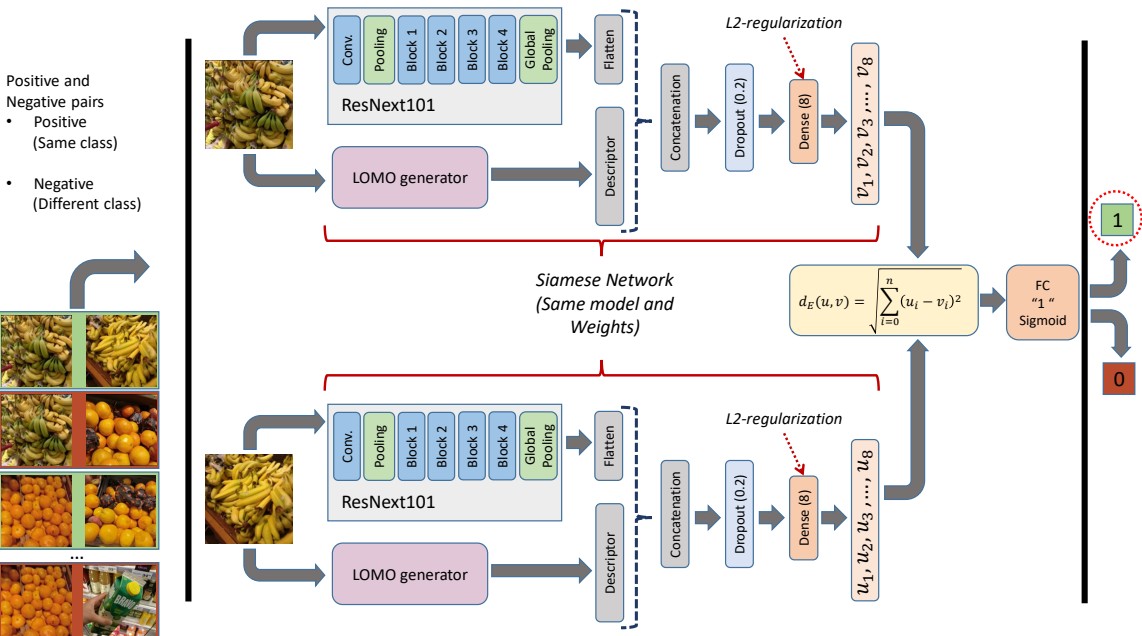

**Figure 6.** Third approach: Siamese net with ResNeXt-101-32x8d and LOMO descriptor.

## 4. Experiments and Results Discussion

Two different sorts of experiments have been carried out. The first experiment (Section 4.1) consisted of improving the baseline classification set by the authors of the Grocery Store Dataset [3]. The second experiment (Section 4.2) has consisted in integrating multiple classifiers into a Siamese network to show how this method improves the current OSL techniques. Section 4.3 shows the Results Discussion.

### 4.1. Improving the Classification Base Line

The authors of the Grocery Store Dataset [3] trained a DenseNet-169, connecting the feature vector to an SVM classifier. They obtained 85% accuracy on test by using a model on which they performed a fine tuning. In the same manner, they tested a DenseNet-169 classification network without using SVM. In this case, they obtained 84% accuracy.

In our experiment, two recent models have been evaluated: a ResNeXt-101 with cardinality 32 and a bottleneck width of eight [4] and a ResNet-152 [33]. The ResNeXt-101 model has been modified by readapting the input shape to $600 \times 600 \times 3$. We have divided the training set, composed of 2640 images, into 1820 images for training (68.94%) and 820 for validation (31.06%). The reason for this distribution is that we have balanced the distribution, to try to distribute 30% of the images of each category to validation in a random manner. The test set consists of 2485 images, and it has been pre-defined by the authors of the Grocery Store Dataset [3].

The training of the ResNeXt-101 has been performed by using an Adam optimizer with a learning rate of 0.00005. The feature vector was connected to an output *dense* layer of 81 categories with a *softmax* activation function. The initial weights used were those of ImageNet [31]. A batch size of eight was used, and the model was trained for 40 epochs, reaching the highest validation accuracy at epoch 30. The training accuracy was 99.78%, while the validation accuracy was 99.76% in epoch 30. The evaluation of the model was performed by using the test set. The test accuracy was 90.80%, with a precision of 92.50%, recall of 92.10%, balanced accuracy of 92.09% and F1-Score of 92.30%. The same experiment was carried out with a ResNet-152. However, despite the depth of this network, the results were slightly worse. The test accuracy was 89.90%, with a precision of 91.60%, recall of 91.10%, balanced accuracy of 91.09% and F1-Score of 91.30%.

The next experiment consisted in adding a dropout layer with a random deactivation of 20% to the input of the last classification layer of the ResNeXt-101. This modification of the model improved the result up to 91.60% test accuracy, 93.60% precision, 92.20% recall, 92.18% balanced accuracy and 92.18% F1-Score. Figure 7a shows the training and validation accuracy, while Figure 7b shows the loss of this model.

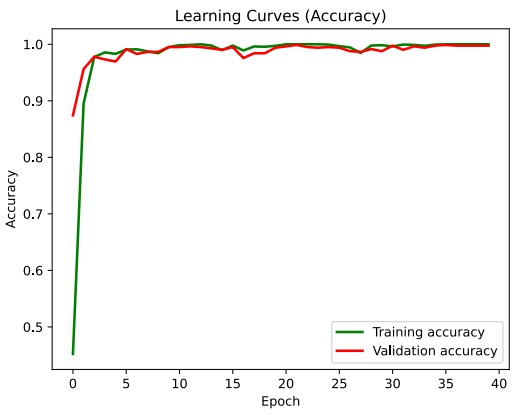
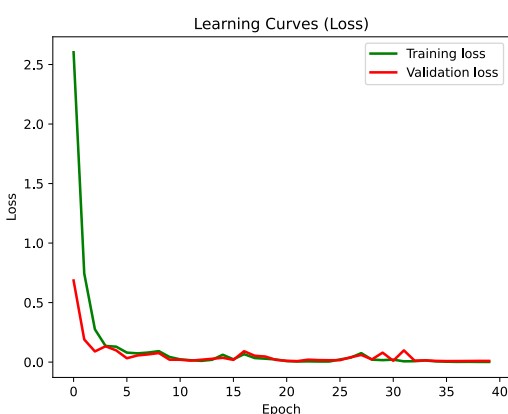

(**a**) ResNeXt-101 training/validation accuracy with 20% dropout.

(**b**) ResNeXt-101 training/validation loss with 20% dropout.

**Figure 7.** ResNeXt-101 training with 20% dropout.

A Precision-Recall curve with the average precision score micro-averaged over all classes is displayed in Figure 8. In addition, a Multi-Class Precision-Recall curve with all classes is displayed in Figure 9. The evaluation of the model shows a good performance, resulting in the 96% AUC (Area Under the Curve).

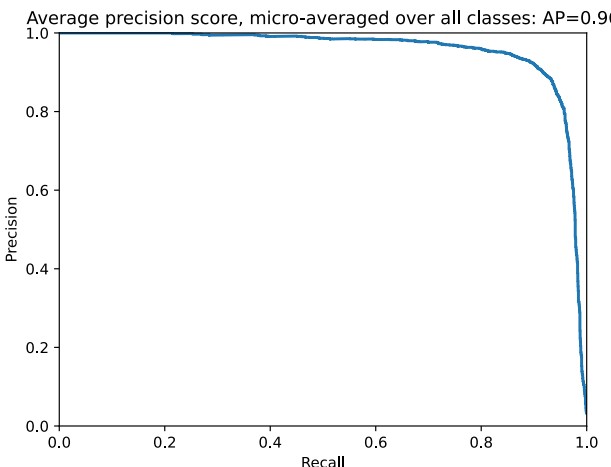

**Figure 8.** Precision-Recall curve with the average precision score micro-averaged over all classes.

The confusion matrix of this model is shown in Table 1. For the sake of space, instead of showing the 81 fine classes, we show the 43 coarse categories where the results have been grouped. Although the overall results have improved, the confusion matrix shows two categories that the model is not able to detect properly: oranges and satsumas.

In order to avoid confusion between oranges and satsumas, another model has been trained to distinguish between oranges and satsumas, again with a ResNeXt-101 with a 20% dropout layer. When the result of the first model is oranges or satsumas, the image is sent to the second classifier. The training/validation and test sets are the same as those used for the first model but only filtering these two classes. The training result of this second model has been 98.86% training accuracy and 94.87% validation accuracy. The test

accuracy in the detection of oranges/satsumas has been 79%, 79% precision, 78.5% recall, 78.5% balanced accuracy and 78.8% F1-Score.

By staggering the two models, the test accuracy has increased to 92%. Table 2 shows the summary of the results. Our model significantly improves the results obtained and established as a baseline in the Grocery Store Dataset [3] article, improving from 85% in the model they trained with a tuned DenseNet-152 and an SVM classifier to 92%.

A Precision-Recall curve for the tiered model with the average precision score, micro-averaged over all classes, is displayed in Figure 10. In addition, a Multi-Class Precision-Recall curve with all classes is displayed in Figure 11. The evaluation of the model shows good performance, resulting in the 96% AUC (Area Under the Curve).

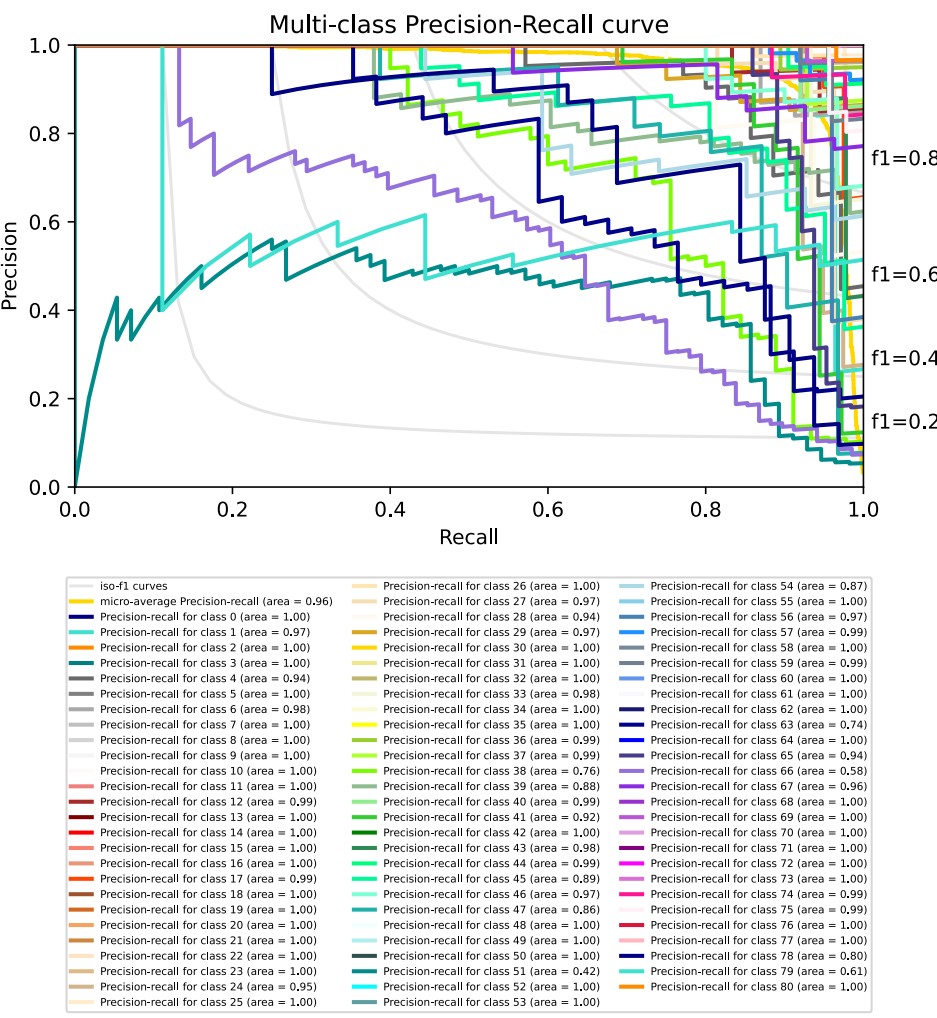

**Figure 9.** Multi-Class Precision-Recall curve.

**Table 1.** Confusion Matrix of the ResNeXt-101 model with 20% dropout.

| Actual \ Predicted | Apple | Asparagus | Aubergine | Avocado | Banana | Brown-Cap-Mushroom | Cabbage | Carrots | Cucumber | Garlic | Ginger | Juice | Kiwi | Leek | Lemon | Lime | Mango | Melon | Milk | Nectarine | Oat-Milk | Oatghurt | Onion | Orange | Papaya | Passion-Fruit | Peach | Pear | Pepper | Pineapple | Plum | Pomegranate | Potato | Red-Beet | Red-Grapefruit | Satsumas | Sour-Cream | Sour-Milk | Soy-Milk | Soyghurt | Tomato | Yoghurt | Zucchini |
|---|---|---|---|---|---|---|---|---|---|---|---|---|---|---|---|---|---|---|---|---|---|---|---|---|---|---|---|---|---|---|---|---|---|---|---|---|---|---|---|---|---|---|---|
| Apple | 0.99 | 0.0 | 0.0 | 0.0 | 0.0 | 0.0 | 0.0 | 0.0 | 0.0 | 0.0 | 0.0 | 0.0 | 0.0 | 0.0 | 0.0 | 0.0 | 0.0 | 0.0 | 0.0 | 0.0 | 0.0 | 0.0 | 0.0 | 0.0 | 0.0 | 0.0 | 0.0 | 0.01 | 0.0 | 0.0 | 0.0 | 0.0 | 0.0 | 0.0 | 0.0 | 0.0 | 0.0 | 0.0 | 0.0 | 0.0 | 0.0 | 0.0 | 0.0 |
| Asparagus | 0.0 | 1.0 | 0.0 | 0.0 | 0.0 | 0.0 | 0.0 | 0.0 | 0.0 | 0.0 | 0.0 | 0.0 | 0.0 | 0.0 | 0.0 | 0.0 | 0.0 | 0.0 | 0.0 | 0.0 | 0.0 | 0.0 | 0.0 | 0.0 | 0.0 | 0.0 | 0.0 | 0.0 | 0.0 | 0.0 | 0.0 | 0.0 | 0.0 | 0.0 | 0.0 | 0.0 | 0.0 | 0.0 | 0.0 | 0.0 | 0.0 | 0.0 | 0.0 |
| Aubergine | 0.0 | 0.0 | 1.0 | 0.0 | 0.0 | 0.0 | 0.0 | 0.0 | 0.0 | 0.0 | 0.0 | 0.0 | 0.0 | 0.0 | 0.0 | 0.0 | 0.0 | 0.0 | 0.0 | 0.0 | 0.0 | 0.0 | 0.0 | 0.0 | 0.0 | 0.0 | 0.0 | 0.0 | 0.0 | 0.0 | 0.0 | 0.0 | 0.0 | 0.0 | 0.0 | 0.0 | 0.0 | 0.0 | 0.0 | 0.0 | 0.0 | 0.0 | 0.0 |
| Avocado | 0.0 | 0.0 | 0.0 | 0.92 | 0.0 | 0.0 | 0.0 | 0.0 | 0.0 | 0.0 | 0.0 | 0.0 | 0.0 | 0.0 | 0.0 | 0.0 | 0.0 | 0.0 | 0.0 | 0.0 | 0.0 | 0.0 | 0.0 | 0.05 | 0.0 | 0.0 | 0.0 | 0.0 | 0.0 | 0.0 | 0.0 | 0.0 | 0.02 | 0.0 | 0.0 | 0.0 | 0.0 | 0.0 | 0.0 | 0.0 | 0.0 | 0.0 | 0.0 |
| Banana | 0.0 | 0.0 | 0.0 | 0.0 | 1.0 | 0.0 | 0.0 | 0.0 | 0.0 | 0.0 | 0.0 | 0.0 | 0.0 | 0.0 | 0.0 | 0.0 | 0.0 | 0.0 | 0.0 | 0.0 | 0.0 | 0.0 | 0.0 | 0.0 | 0.0 | 0.0 | 0.0 | 0.0 | 0.0 | 0.0 | 0.0 | 0.0 | 0.0 | 0.0 | 0.0 | 0.0 | 0.0 | 0.0 | 0.0 | 0.0 | 0.0 | 0.0 | 0.0 |
| Brown-Cap-Mushroom | 0.0 | 0.0 | 0.0 | 0.0 | 0.0 | 1.0 | 0.0 | 0.0 | 0.0 | 0.0 | 0.0 | 0.0 | 0.0 | 0.0 | 0.0 | 0.0 | 0.0 | 0.0 | 0.0 | 0.0 | 0.0 | 0.0 | 0.0 | 0.0 | 0.0 | 0.0 | 0.0 | 0.0 | 0.0 | 0.0 | 0.0 | 0.0 | 0.0 | 0.0 | 0.0 | 0.0 | 0.0 | 0.0 | 0.0 | 0.0 | 0.0 | 0.0 | 0.0 |
| Cabbage | 0.0 | 0.0 | 0.0 | 0.0 | 0.0 | 0.0 | 1.0 | 0.0 | 0.0 | 0.0 | 0.0 | 0.0 | 0.0 | 0.0 | 0.0 | 0.0 | 0.0 | 0.0 | 0.0 | 0.0 | 0.0 | 0.0 | 0.0 | 0.0 | 0.0 | 0.0 | 0.0 | 0.0 | 0.0 | 0.0 | 0.0 | 0.0 | 0.0 | 0.0 | 0.0 | 0.0 | 0.0 | 0.0 | 0.0 | 0.0 | 0.0 | 0.0 | 0.0 |
| Carrots | 0.0 | 0.0 | 0.0 | 0.0 | 0.0 | 0.0 | 0.0 | 0.98 | 0.0 | 0.0 | 0.0 | 0.0 | 0.0 | 0.0 | 0.0 | 0.0 | 0.0 | 0.0 | 0.0 | 0.0 | 0.0 | 0.0 | 0.0 | 0.0 | 0.0 | 0.0 | 0.0 | 0.0 | 0.0 | 0.0 | 0.0 | 0.0 | 0.02 | 0.0 | 0.0 | 0.0 | 0.0 | 0.0 | 0.0 | 0.0 | 0.0 | 0.0 | 0.0 |
| Cucumber | 0.04 | 0.0 | 0.0 | 0.0 | 0.0 | 0.0 | 0.0 | 0.0 | 0.85 | 0.0 | 0.0 | 0.0 | 0.0 | 0.0 | 0.0 | 0.0 | 0.0 | 0.0 | 0.0 | 0.0 | 0.0 | 0.0 | 0.0 | 0.0 | 0.0 | 0.0 | 0.0 | 0.0 | 0.0 | 0.0 | 0.0 | 0.0 | 0.0 | 0.0 | 0.0 | 0.0 | 0.0 | 0.0 | 0.0 | 0.0 | 0.0 | 0.04 | 0.07 |
| Garlic | 0.0 | 0.0 | 0.0 | 0.0 | 0.0 | 0.0 | 0.0 | 0.0 | 0.0 | 1.0 | 0.0 | 0.0 | 0.0 | 0.0 | 0.0 | 0.0 | 0.0 | 0.0 | 0.0 | 0.0 | 0.0 | 0.0 | 0.0 | 0.0 | 0.0 | 0.0 | 0.0 | 0.0 | 0.0 | 0.0 | 0.0 | 0.0 | 0.0 | 0.0 | 0.0 | 0.0 | 0.0 | 0.0 | 0.0 | 0.0 | 0.0 | 0.0 | 0.0 |
| Ginger | 0.0 | 0.0 | 0.0 | 0.0 | 0.0 | 0.0 | 0.0 | 0.0 | 0.0 | 0.07 | 0.87 | 0.0 | 0.0 | 0.0 | 0.0 | 0.0 | 0.0 | 0.0 | 0.0 | 0.0 | 0.0 | 0.0 | 0.0 | 0.0 | 0.0 | 0.0 | 0.0 | 0.07 | 0.0 | 0.0 | 0.0 | 0.0 | 0.0 | 0.0 | 0.0 | 0.0 | 0.0 | 0.0 | 0.0 | 0.0 | 0.0 | 0.0 | 0.0 |
| Juice | 0.0 | 0.0 | 0.0 | 0.0 | 0.0 | 0.0 | 0.0 | 0.0 | 0.0 | 0.0 | 0.0 | 1.0 | 0.0 | 0.0 | 0.0 | 0.0 | 0.0 | 0.0 | 0.0 | 0.0 | 0.0 | 0.0 | 0.0 | 0.0 | 0.0 | 0.0 | 0.0 | 0.0 | 0.0 | 0.0 | 0.0 | 0.0 | 0.0 | 0.0 | 0.0 | 0.0 | 0.0 | 0.0 | 0.0 | 0.0 | 0.0 | 0.0 | 0.0 |
| Kiwi | 0.0 | 0.0 | 0.0 | 0.0 | 0.0 | 0.0 | 0.0 | 0.0 | 0.0 | 0.0 | 0.0 | 0.0 | 0.93 | 0.0 | 0.0 | 0.0 | 0.0 | 0.0 | 0.0 | 0.0 | 0.0 | 0.0 | 0.0 | 0.07 | 0.0 | 0.0 | 0.0 | 0.0 | 0.0 | 0.0 | 0.0 | 0.0 | 0.0 | 0.0 | 0.0 | 0.0 | 0.0 | 0.0 | 0.0 | 0.0 | 0.0 | 0.0 | 0.0 |
| Leek | 0.0 | 0.0 | 0.0 | 0.0 | 0.0 | 0.0 | 0.0 | 0.0 | 0.0 | 0.0 | 0.0 | 0.0 | 0.0 | 0.9 | 0.0 | 0.0 | 0.0 | 0.0 | 0.0 | 0.0 | 0.0 | 0.0 | 0.0 | 0.0 | 0.0 | 0.0 | 0.0 | 0.1 | 0.0 | 0.0 | 0.0 | 0.0 | 0.0 | 0.0 | 0.0 | 0.0 | 0.0 | 0.0 | 0.0 | 0.0 | 0.0 | 0.0 | 0.0 |
| Lemon | 0.05 | 0.0 | 0.0 | 0.0 | 0.0 | 0.0 | 0.0 | 0.0 | 0.0 | 0.0 | 0.0 | 0.0 | 0.0 | 0.0 | 0.59 | 0.05 | 0.0 | 0.0 | 0.0 | 0.0 | 0.0 | 0.0 | 0.0 | 0.05 | 0.0 | 0.0 | 0.0 | 0.17 | 0.0 | 0.1 | 0.0 | 0.0 | 0.0 | 0.0 | 0.0 | 0.0 | 0.0 | 0.0 | 0.0 | 0.0 | 0.0 | 0.0 | 0.0 |
| Lime | 0.07 | 0.0 | 0.0 | 0.0 | 0.0 | 0.0 | 0.0 | 0.0 | 0.0 | 0.0 | 0.0 | 0.0 | 0.0 | 0.0 | 0.0 | 0.8 | 0.07 | 0.0 | 0.0 | 0.0 | 0.0 | 0.0 | 0.0 | 0.0 | 0.0 | 0.0 | 0.0 | 0.07 | 0.0 | 0.0 | 0.0 | 0.0 | 0.0 | 0.0 | 0.0 | 0.0 | 0.0 | 0.0 | 0.0 | 0.0 | 0.0 | 0.0 | 0.0 |
| Mango | 0.19 | 0.0 | 0.0 | 0.0 | 0.0 | 0.0 | 0.0 | 0.0 | 0.0 | 0.0 | 0.0 | 0.0 | 0.0 | 0.0 | 0.0 | 0.03 | 0.71 | 0.0 | 0.0 | 0.0 | 0.0 | 0.0 | 0.0 | 0.03 | 0.0 | 0.0 | 0.0 | 0.03 | 0.0 | 0.0 | 0.0 | 0.0 | 0.0 | 0.0 | 0.0 | 0.0 | 0.0 | 0.0 | 0.0 | 0.0 | 0.0 | 0.0 | 0.0 |
| Melon | 0.0 | 0.0 | 0.0 | 0.0 | 0.0 | 0.0 | 0.0 | 0.0 | 0.0 | 0.0 | 0.0 | 0.01 | 0.0 | 0.0 | 0.0 | 0.0 | 0.01 | 0.93 | 0.0 | 0.0 | 0.0 | 0.0 | 0.0 | 0.05 | 0.0 | 0.0 | 0.0 | 0.0 | 0.0 | 0.0 | 0.0 | 0.0 | 0.0 | 0.0 | 0.0 | 0.01 | 0.0 | 0.0 | 0.0 | 0.0 | 0.0 | 0.0 | 0.0 |
| Milk | 0.0 | 0.0 | 0.0 | 0.0 | 0.0 | 0.0 | 0.0 | 0.0 | 0.0 | 0.0 | 0.0 | 0.0 | 0.0 | 0.0 | 0.0 | 0.0 | 0.0 | 0.0 | 1.0 | 0.0 | 0.0 | 0.0 | 0.0 | 0.0 | 0.0 | 0.0 | 0.0 | 0.0 | 0.0 | 0.0 | 0.0 | 0.0 | 0.0 | 0.0 | 0.0 | 0.0 | 0.0 | 0.0 | 0.0 | 0.0 | 0.0 | 0.0 | 0.0 |
| Nectarine | 0.0 | 0.0 | 0.0 | 0.0 | 0.0 | 0.0 | 0.0 | 0.0 | 0.0 | 0.0 | 0.0 | 0.0 | 0.0 | 0.0 | 0.0 | 0.0 | 0.0 | 0.0 | 0.0 | 1.0 | 0.0 | 0.0 | 0.0 | 0.0 | 0.0 | 0.0 | 0.0 | 0.0 | 0.0 | 0.0 | 0.0 | 0.0 | 0.0 | 0.0 | 0.0 | 0.0 | 0.0 | 0.0 | 0.0 | 0.0 | 0.0 | 0.0 | 0.0 |
| Oat-Milk | 0.0 | 0.0 | 0.0 | 0.0 | 0.0 | 0.0 | 0.0 | 0.0 | 0.0 | 0.0 | 0.0 | 0.0 | 0.0 | 0.0 | 0.0 | 0.0 | 0.0 | 0.0 | 0.0 | 0.0 | 1.0 | 0.0 | 0.0 | 0.0 | 0.0 | 0.0 | 0.0 | 0.0 | 0.0 | 0.0 | 0.0 | 0.0 | 0.0 | 0.0 | 0.0 | 0.0 | 0.0 | 0.0 | 0.0 | 0.0 | 0.0 | 0.0 | 0.0 |
| Oatghurt | 0.0 | 0.0 | 0.0 | 0.0 | 0.0 | 0.0 | 0.0 | 0.0 | 0.0 | 0.0 | 0.0 | 0.0 | 0.0 | 0.0 | 0.0 | 0.0 | 0.0 | 0.0 | 0.0 | 0.0 | 0.0 | 1.0 | 0.0 | 0.0 | 0.0 | 0.0 | 0.0 | 0.0 | 0.0 | 0.0 | 0.0 | 0.0 | 0.0 | 0.0 | 0.0 | 0.0 | 0.0 | 0.0 | 0.0 | 0.0 | 0.0 | 0.0 | 0.0 |
| Onion | 0.0 | 0.0 | 0.0 | 0.0 | 0.0 | 0.0 | 0.0 | 0.0 | 0.0 | 0.0 | 0.0 | 0.0 | 0.0 | 0.0 | 0.0 | 0.0 | 0.0 | 0.0 | 0.0 | 0.0 | 0.0 | 0.0 | 1.0 | 0.0 | 0.0 | 0.0 | 0.0 | 0.0 | 0.0 | 0.0 | 0.0 | 0.0 | 0.0 | 0.0 | 0.0 | 0.0 | 0.0 | 0.0 | 0.0 | 0.0 | 0.0 | 0.0 | 0.0 |
| Orange | 0.0 | 0.0 | 0.0 | 0.0 | 0.0 | 0.0 | 0.0 | 0.0 | 0.0 | 0.0 | 0.0 | 0.0 | 0.0 | 0.0 | 0.0 | 0.0 | 0.0 | 0.0 | 0.0 | 0.0 | 0.0 | 0.0 | 0.0 | 0.64 | 0.0 | 0.0 | 0.0 | 0.0 | 0.0 | 0.0 | 0.0 | 0.0 | 0.0 | 0.0 | 0.09 | 0.27 | 0.0 | 0.0 | 0.0 | 0.0 | 0.0 | 0.0 | 0.0 |
| Papaya | 0.0 | 0.0 | 0.0 | 0.0 | 0.0 | 0.0 | 0.0 | 0.0 | 0.0 | 0.0 | 0.0 | 0.0 | 0.0 | 0.0 | 0.0 | 0.0 | 0.0 | 0.0 | 0.0 | 0.0 | 0.0 | 0.0 | 0.0 | 0.0 | 1.0 | 0.0 | 0.0 | 0.0 | 0.0 | 0.0 | 0.0 | 0.0 | 0.0 | 0.0 | 0.0 | 0.0 | 0.0 | 0.0 | 0.0 | 0.0 | 0.0 | 0.0 | 0.0 |
| Passion-Fruit | 0.0 | 0.0 | 0.22 | 0.0 | 0.0 | 0.0 | 0.0 | 0.0 | 0.0 | 0.0 | 0.0 | 0.0 | 0.0 | 0.0 | 0.0 | 0.0 | 0.04 | 0.0 | 0.0 | 0.0 | 0.0 | 0.0 | 0.0 | 0.0 | 0.0 | 0.63 | 0.0 | 0.0 | 0.0 | 0.11 | 0.0 | 0.0 | 0.0 | 0.0 | 0.0 | 0.0 | 0.0 | 0.0 | 0.0 | 0.0 | 0.0 | 0.0 | 0.0 |
| Peach | 0.0 | 0.0 | 0.0 | 0.0 | 0.0 | 0.0 | 0.0 | 0.0 | 0.0 | 0.0 | 0.0 | 0.0 | 0.0 | 0.0 | 0.0 | 0.0 | 0.0 | 0.0 | 0.0 | 0.0 | 0.0 | 0.0 | 0.0 | 0.0 | 0.0 | 0.0 | 1.0 | 0.0 | 0.0 | 0.0 | 0.0 | 0.0 | 0.0 | 0.0 | 0.0 | 0.0 | 0.0 | 0.0 | 0.0 | 0.0 | 0.0 | 0.0 | 0.0 |
| Pear | 0.04 | 0.0 | 0.0 | 0.0 | 0.01 | 0.0 | 0.0 | 0.0 | 0.0 | 0.0 | 0.0 | 0.0 | 0.0 | 0.0 | 0.0 | 0.0 | 0.01 | 0.0 | 0.0 | 0.0 | 0.0 | 0.0 | 0.0 | 0.01 | 0.0 | 0.0 | 0.0 | 0.94 | 0.0 | 0.0 | 0.0 | 0.0 | 0.0 | 0.0 | 0.0 | 0.0 | 0.0 | 0.0 | 0.0 | 0.0 | 0.01 | 0.0 | 0.0 |
| Pepper | 0.0 | 0.0 | 0.0 | 0.0 | 0.0 | 0.0 | 0.0 | 0.0 | 0.0 | 0.0 | 0.0 | 0.0 | 0.0 | 0.0 | 0.0 | 0.0 | 0.0 | 0.0 | 0.0 | 0.0 | 0.0 | 0.0 | 0.0 | 0.0 | 0.0 | 0.0 | 0.0 | 0.0 | 0.99 | 0.0 | 0.0 | 0.0 | 0.0 | 0.0 | 0.0 | 0.0 | 0.0 | 0.0 | 0.0 | 0.0 | 0.01 | 0.0 | 0.0 |
| Pineapple | 0.0 | 0.0 | 0.0 | 0.0 | 0.0 | 0.0 | 0.0 | 0.0 | 0.0 | 0.0 | 0.0 | 0.04 | 0.0 | 0.0 | 0.0 | 0.0 | 0.0 | 0.0 | 0.0 | 0.0 | 0.0 | 0.0 | 0.0 | 0.0 | 0.0 | 0.0 | 0.0 | 0.0 | 0.0 | 0.96 | 0.0 | 0.0 | 0.0 | 0.0 | 0.0 | 0.0 | 0.0 | 0.0 | 0.0 | 0.0 | 0.0 | 0.0 | 0.0 |
| Plum | 0.0 | 0.0 | 0.0 | 0.0 | 0.0 | 0.0 | 0.0 | 0.0 | 0.0 | 0.0 | 0.0 | 0.0 | 0.0 | 0.0 | 0.0 | 0.0 | 0.0 | 0.0 | 0.0 | 0.0 | 0.0 | 0.0 | 0.0 | 0.0 | 0.0 | 0.0 | 0.0 | 0.0 | 0.0 | 0.0 | 1.0 | 0.0 | 0.0 | 0.0 | 0.0 | 0.0 | 0.0 | 0.0 | 0.0 | 0.0 | 0.0 | 0.0 | 0.0 |
| Pomegranate | 0.0 | 0.0 | 0.0 | 0.0 | 0.0 | 0.0 | 0.0 | 0.0 | 0.0 | 0.0 | 0.0 | 0.0 | 0.0 | 0.0 | 0.0 | 0.0 | 0.04 | 0.0 | 0.0 | 0.0 | 0.0 | 0.0 | 0.0 | 0.04 | 0.0 | 0.0 | 0.0 | 0.0 | 0.0 | 0.0 | 0.0 | 0.92 | 0.0 | 0.0 | 0.0 | 0.0 | 0.0 | 0.0 | 0.0 | 0.0 | 0.0 | 0.0 | 0.0 |
| Potato | 0.0 | 0.0 | 0.0 | 0.0 | 0.0 | 0.0 | 0.0 | 0.0 | 0.0 | 0.0 | 0.0 | 0.0 | 0.0 | 0.0 | 0.0 | 0.0 | 0.0 | 0.0 | 0.0 | 0.0 | 0.0 | 0.0 | 0.0 | 0.0 | 0.0 | 0.0 | 0.0 | 0.0 | 0.0 | 0.0 | 0.0 | 0.0 | 1.0 | 0.0 | 0.0 | 0.0 | 0.0 | 0.0 | 0.0 | 0.0 | 0.0 | 0.0 | 0.0 |
| Red-Beet | 0.0 | 0.0 | 0.0 | 0.0 | 0.0 | 0.0 | 0.0 | 0.0 | 0.0 | 0.0 | 0.0 | 0.0 | 0.0 | 0.0 | 0.0 | 0.0 | 0.0 | 0.0 | 0.0 | 0.0 | 0.0 | 0.0 | 0.0 | 0.0 | 0.0 | 0.0 | 0.0 | 0.0 | 0.0 | 0.0 | 0.0 | 0.0 | 0.0 | 1.0 | 0.0 | 0.0 | 0.0 | 0.0 | 0.0 | 0.0 | 0.0 | 0.0 | 0.0 |
| Red-Grapefruit | 0.0 | 0.0 | 0.0 | 0.0 | 0.0 | 0.0 | 0.0 | 0.0 | 0.0 | 0.0 | 0.0 | 0.0 | 0.0 | 0.0 | 0.0 | 0.0 | 0.03 | 0.0 | 0.0 | 0.0 | 0.0 | 0.0 | 0.0 | 0.26 | 0.0 | 0.0 | 0.0 | 0.0 | 0.0 | 0.0 | 0.0 | 0.0 | 0.0 | 0.0 | 0.62 | 0.09 | 0.0 | 0.0 | 0.0 | 0.0 | 0.0 | 0.0 | 0.0 |
| Satsumas | 0.0 | 0.0 | 0.0 | 0.0 | 0.0 | 0.0 | 0.0 | 0.0 | 0.0 | 0.0 | 0.0 | 0.0 | 0.0 | 0.0 | 0.0 | 0.0 | 0.0 | 0.0 | 0.0 | 0.0 | 0.0 | 0.0 | 0.0 | 0.46 | 0.0 | 0.0 | 0.0 | 0.0 | 0.0 | 0.0 | 0.0 | 0.0 | 0.0 | 0.0 | 0.0 | 0.54 | 0.0 | 0.0 | 0.0 | 0.0 | 0.0 | 0.0 | 0.0 |
| Sour-Cream | 0.0 | 0.0 | 0.0 | 0.0 | 0.0 | 0.0 | 0.0 | 0.0 | 0.0 | 0.0 | 0.0 | 0.0 | 0.0 | 0.0 | 0.0 | 0.0 | 0.0 | 0.0 | 0.02 | 0.0 | 0.0 | 0.0 | 0.0 | 0.0 | 0.0 | 0.0 | 0.0 | 0.0 | 0.0 | 0.0 | 0.0 | 0.0 | 0.0 | 0.0 | 0.0 | 0.0 | 0.93 | 0.05 | 0.0 | 0.0 | 0.0 | 0.0 | 0.0 |
| Sour-Milk | 0.0 | 0.0 | 0.0 | 0.0 | 0.0 | 0.0 | 0.0 | 0.0 | 0.0 | 0.0 | 0.0 | 0.0 | 0.0 | 0.0 | 0.0 | 0.0 | 0.0 | 0.0 | 0.0 | 0.0 | 0.0 | 0.0 | 0.0 | 0.0 | 0.0 | 0.0 | 0.0 | 0.0 | 0.0 | 0.0 | 0.0 | 0.0 | 0.0 | 0.0 | 0.0 | 0.0 | 0.0 | 1.0 | 0.0 | 0.0 | 0.0 | 0.0 | 0.0 |
| Soy-Milk | 0.0 | 0.0 | 0.0 | 0.0 | 0.0 | 0.0 | 0.0 | 0.0 | 0.0 | 0.0 | 0.0 | 0.0 | 0.0 | 0.0 | 0.0 | 0.0 | 0.0 | 0.0 | 0.0 | 0.0 | 0.0 | 0.0 | 0.0 | 0.0 | 0.0 | 0.0 | 0.0 | 0.0 | 0.0 | 0.0 | 0.0 | 0.0 | 0.0 | 0.0 | 0.0 | 0.0 | 0.0 | 0.0 | 1.0 | 0.0 | 0.0 | 0.0 | 0.0 |
| Soyghurt | 0.0 | 0.0 | 0.0 | 0.0 | 0.0 | 0.0 | 0.0 | 0.0 | 0.0 | 0.0 | 0.0 | 0.0 | 0.0 | 0.0 | 0.0 | 0.0 | 0.0 | 0.0 | 0.0 | 0.0 | 0.0 | 0.0 | 0.0 | 0.0 | 0.0 | 0.0 | 0.0 | 0.0 | 0.0 | 0.0 | 0.0 | 0.0 | 0.0 | 0.0 | 0.0 | 0.0 | 0.0 | 0.0 | 0.0 | 1.0 | 0.0 | 0.0 | 0.0 |
| Tomato | 0.0 | 0.0 | 0.0 | 0.0 | 0.0 | 0.0 | 0.0 | 0.0 | 0.0 | 0.0 | 0.0 | 0.0 | 0.0 | 0.0 | 0.0 | 0.0 | 0.0 | 0.0 | 0.0 | 0.0 | 0.0 | 0.0 | 0.0 | 0.0 | 0.0 | 0.0 | 0.0 | 0.0 | 0.0 | 0.0 | 0.0 | 0.0 | 0.0 | 0.0 | 0.0 | 0.0 | 0.0 | 0.0 | 0.0 | 0.0 | 1.0 | 0.0 | 0.0 |
| Yoghurt | 0.0 | 0.0 | 0.0 | 0.0 | 0.0 | 0.0 | 0.0 | 0.0 | 0.0 | 0.0 | 0.0 | 0.0 | 0.0 | 0.0 | 0.0 | 0.0 | 0.0 | 0.0 | 0.0 | 0.0 | 0.0 | 0.0 | 0.0 | 0.0 | 0.0 | 0.0 | 0.0 | 0.0 | 0.0 | 0.0 | 0.0 | 0.0 | 0.0 | 0.0 | 0.0 | 0.0 | 0.0 | 0.0 | 0.0 | 0.0 | 0.0 | 1.0 | 0.0 |
| Zucchini | 0.0 | 0.0 | 0.0 | 0.0 | 0.0 | 0.0 | 0.0 | 0.0 | 0.0 | 0.0 | 0.0 | 0.0 | 0.0 | 0.0 | 0.0 | 0.0 | 0.0 | 0.0 | 0.0 | 0.0 | 0.0 | 0.0 | 0.0 | 0.0 | 0.0 | 0.0 | 0.0 | 0.0 | 0.0 | 0.0 | 0.0 | 0.0 | 0.0 | 0.0 | 0.0 | 0.0 | 0.0 | 0.0 | 0.0 | 0.0 | 0.0 | 0.0 | 1.0 |

**Table 2.** Comparison of different classification models (% except for loss).

| Metric | ResNeXt-101 without *dropout* | ResNeXt-101 with *dropout* | Tiered Model ResNeXt-101 with *dropout* | ResNet-152 | DenseNet-169 with SVM [3] |
|---|---|---|---|---|---|
| **Training accuracy** | 99.78 | 1.00 | 1.00/1.00 | 1.00 | |
| **Training loss** | 0.0149 | 0.0047 | 0.0047/0.0044 | 0.0062 | |
| **Validation accuracy** | 99.76 | 99.88 | 99.88/1.00 | 99.63 | |
| **Validation loss** | 0.0139 | 0.0082 | 0.0082/0.0089 | 0.0169 | |
| **Test accuracy** | 90.80 | 91.60 | 92.00 | 89.90 | 85.00 |
| **Balanced test accuracy** | 92.09 | 92.18 | 93.06 | 91.09 | |
| **Precision** | 92.50 | 93.60 | 93.50 | 91.60 | |
| **Recall** | 92.10 | 92.20 | 93.10 | 91.10 | |
| **F1-SCORE** | 92.30 | 92.18 | 93.30 | 91.30 | |

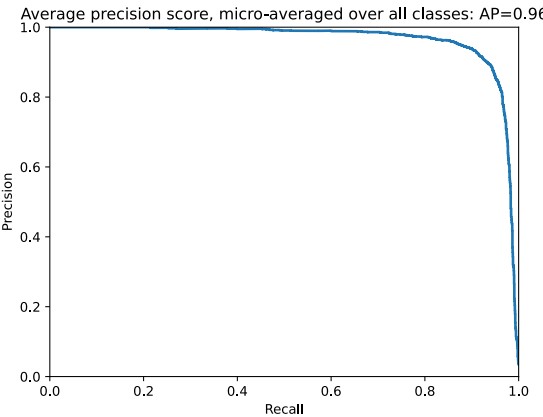

**Figure 10.** Precision-Recall curve for the tiered model with the average precision score micro-averaged over all classes.

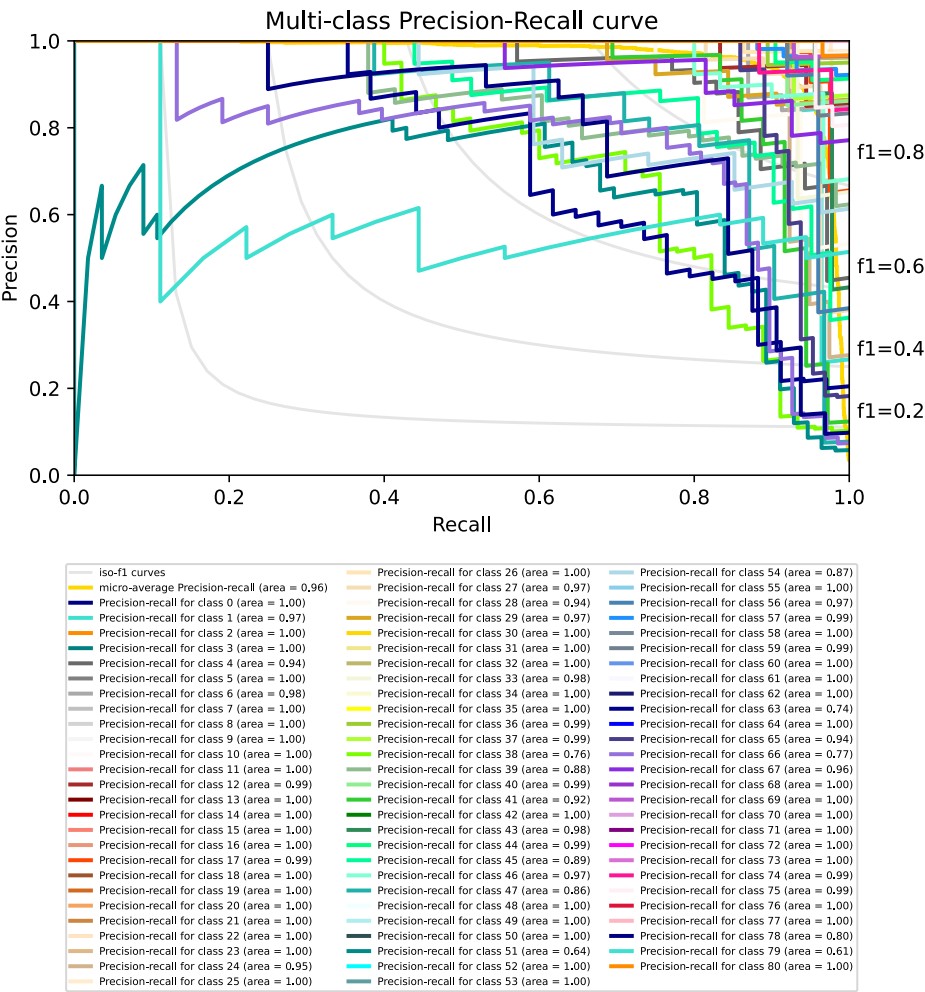

**Figure 11.** Multi-Class Precision-Recall curve for the tiered model.

As observed in the confusion matrix of this tiered model (Table 3), the detection of the oranges and satsumas has been improved.

**Table 3.** Confusion Matrix of the tiered ResNeXt-101 model (with 20% dropout).

| Actual \ Predicted | Apple | Asparagus | Aubergine | Avocado | Banana | Brown-Cap-Mushroom | Cabbage | Carrots | Cucumber | Garlic | Ginger | Juice | Kiwi | Leek | Lemon | Lime | Mango | Melon | Milk | Nectarine | Oat-Milk | Oatghurt | Onion | Orange | Papaya | Passion-Fruit | Peach | Pear | Pepper | Pineapple | Plum | Pomegranate | Potato | Red-Beet | Red-Grapefruit | Satsumas | Sour-Cream | Sour-Milk | Soy-Milk | Soyghurt | Tomato | Yoghurt | Zucchini |
|---|---|---|---|---|---|---|---|---|---|---|---|---|---|---|---|---|---|---|---|---|---|---|---|---|---|---|---|---|---|---|---|---|---|---|---|---|---|---|---|---|---|---|---|
| Apple | 0.99 | 0.0 | 0.0 | 0.0 | 0.0 | 0.0 | 0.0 | 0.0 | 0.0 | 0.0 | 0.0 | 0.0 | 0.0 | 0.0 | 0.0 | 0.0 | 0.0 | 0.0 | 0.0 | 0.0 | 0.0 | 0.0 | 0.0 | 0.0 | 0.0 | 0.0 | 0.0 | 0.01 | 0.0 | 0.0 | 0.0 | 0.0 | 0.0 | 0.0 | 0.0 | 0.0 | 0.0 | 0.0 | 0.0 | 0.0 | 0.0 | 0.0 | 0.0 |
| Asparagus | 0.0 | 1.0 | 0.0 | 0.0 | 0.0 | 0.0 | 0.0 | 0.0 | 0.0 | 0.0 | 0.0 | 0.0 | 0.0 | 0.0 | 0.0 | 0.0 | 0.0 | 0.0 | 0.0 | 0.0 | 0.0 | 0.0 | 0.0 | 0.0 | 0.0 | 0.0 | 0.0 | 0.0 | 0.0 | 0.0 | 0.0 | 0.0 | 0.0 | 0.0 | 0.0 | 0.0 | 0.0 | 0.0 | 0.0 | 0.0 | 0.0 | 0.0 | 0.0 |
| Aubergine | 0.0 | 0.0 | 1.0 | 0.0 | 0.0 | 0.0 | 0.0 | 0.0 | 0.0 | 0.0 | 0.0 | 0.0 | 0.0 | 0.0 | 0.0 | 0.0 | 0.0 | 0.0 | 0.0 | 0.0 | 0.0 | 0.0 | 0.0 | 0.0 | 0.0 | 0.0 | 0.0 | 0.0 | 0.0 | 0.0 | 0.0 | 0.0 | 0.0 | 0.0 | 0.0 | 0.0 | 0.0 | 0.0 | 0.0 | 0.0 | 0.0 | 0.0 | 0.0 |
| Avocado | 0.0 | 0.0 | 0.0 | 0.92 | 0.0 | 0.0 | 0.0 | 0.0 | 0.0 | 0.0 | 0.0 | 0.0 | 0.0 | 0.0 | 0.0 | 0.0 | 0.0 | 0.0 | 0.0 | 0.0 | 0.0 | 0.0 | 0.0 | 0.05 | 0.02 | 0.0 | 0.0 | 0.0 | 0.0 | 0.0 | 0.0 | 0.0 | 0.0 | 0.0 | 0.0 | 0.0 | 0.0 | 0.0 | 0.0 | 0.0 | 0.0 | 0.0 | 0.0 |
| Banana | 0.0 | 0.0 | 0.0 | 0.0 | 1.0 | 0.0 | 0.0 | 0.0 | 0.0 | 0.0 | 0.0 | 0.0 | 0.0 | 0.0 | 0.0 | 0.0 | 0.0 | 0.0 | 0.0 | 0.0 | 0.0 | 0.0 | 0.0 | 0.0 | 0.0 | 0.0 | 0.0 | 0.0 | 0.0 | 0.0 | 0.0 | 0.0 | 0.0 | 0.0 | 0.0 | 0.0 | 0.0 | 0.0 | 0.0 | 0.0 | 0.0 | 0.0 | 0.0 |
| Brown-Cap-Mushroom | 0.0 | 0.0 | 0.0 | 0.0 | 0.0 | 1.0 | 0.0 | 0.0 | 0.0 | 0.0 | 0.0 | 0.0 | 0.0 | 0.0 | 0.0 | 0.0 | 0.0 | 0.0 | 0.0 | 0.0 | 0.0 | 0.0 | 0.0 | 0.0 | 0.0 | 0.0 | 0.0 | 0.0 | 0.0 | 0.0 | 0.0 | 0.0 | 0.0 | 0.0 | 0.0 | 0.0 | 0.0 | 0.0 | 0.0 | 0.0 | 0.0 | 0.0 | 0.0 |
| Cabbage | 0.0 | 0.0 | 0.0 | 0.0 | 0.0 | 0.0 | 1.0 | 0.0 | 0.0 | 0.0 | 0.0 | 0.0 | 0.0 | 0.0 | 0.0 | 0.0 | 0.0 | 0.0 | 0.0 | 0.0 | 0.0 | 0.0 | 0.0 | 0.0 | 0.0 | 0.0 | 0.0 | 0.0 | 0.0 | 0.0 | 0.0 | 0.0 | 0.0 | 0.0 | 0.0 | 0.0 | 0.0 | 0.0 | 0.0 | 0.0 | 0.0 | 0.0 | 0.0 |
| Carrots | 0.0 | 0.0 | 0.0 | 0.0 | 0.0 | 0.0 | 0.0 | 0.98 | 0.0 | 0.0 | 0.0 | 0.0 | 0.0 | 0.0 | 0.0 | 0.0 | 0.0 | 0.0 | 0.0 | 0.0 | 0.0 | 0.0 | 0.0 | 0.0 | 0.0 | 0.0 | 0.0 | 0.0 | 0.0 | 0.0 | 0.0 | 0.0 | 0.02 | 0.0 | 0.0 | 0.0 | 0.0 | 0.0 | 0.0 | 0.0 | 0.0 | 0.0 | 0.0 |
| Cucumber | 0.04 | 0.0 | 0.0 | 0.0 | 0.0 | 0.0 | 0.0 | 0.0 | 0.85 | 0.0 | 0.0 | 0.0 | 0.0 | 0.0 | 0.0 | 0.0 | 0.0 | 0.0 | 0.0 | 0.0 | 0.0 | 0.0 | 0.0 | 0.0 | 0.0 | 0.0 | 0.0 | 0.0 | 0.0 | 0.0 | 0.0 | 0.0 | 0.0 | 0.0 | 0.0 | 0.0 | 0.0 | 0.0 | 0.0 | 0.0 | 0.04 | 0.0 | 0.07 |
| Garlic | 0.0 | 0.0 | 0.0 | 0.0 | 0.0 | 0.0 | 0.0 | 0.0 | 0.0 | 1.0 | 0.0 | 0.0 | 0.0 | 0.0 | 0.0 | 0.0 | 0.0 | 0.0 | 0.0 | 0.0 | 0.0 | 0.0 | 0.0 | 0.0 | 0.0 | 0.0 | 0.0 | 0.0 | 0.0 | 0.0 | 0.0 | 0.0 | 0.0 | 0.0 | 0.0 | 0.0 | 0.0 | 0.0 | 0.0 | 0.0 | 0.0 | 0.0 | 0.0 |
| Ginger | 0.0 | 0.0 | 0.0 | 0.0 | 0.0 | 0.0 | 0.0 | 0.0 | 0.0 | 0.07 | 0.87 | 0.0 | 0.0 | 0.0 | 0.0 | 0.0 | 0.0 | 0.0 | 0.0 | 0.0 | 0.0 | 0.0 | 0.0 | 0.0 | 0.0 | 0.0 | 0.0 | 0.0 | 0.0 | 0.07 | 0.0 | 0.0 | 0.0 | 0.0 | 0.0 | 0.0 | 0.0 | 0.0 | 0.0 | 0.0 | 0.0 | 0.0 | 0.0 |
| Juice | 0.0 | 0.0 | 0.0 | 0.0 | 0.0 | 0.0 | 0.0 | 0.0 | 0.0 | 0.0 | 0.0 | 1.0 | 0.0 | 0.0 | 0.0 | 0.0 | 0.0 | 0.0 | 0.0 | 0.0 | 0.0 | 0.0 | 0.0 | 0.0 | 0.0 | 0.0 | 0.0 | 0.0 | 0.0 | 0.0 | 0.0 | 0.0 | 0.0 | 0.0 | 0.0 | 0.0 | 0.0 | 0.0 | 0.0 | 0.0 | 0.0 | 0.0 | 0.0 |
| Kiwi | 0.0 | 0.0 | 0.0 | 0.0 | 0.0 | 0.0 | 0.0 | 0.0 | 0.0 | 0.0 | 0.0 | 0.0 | 0.93 | 0.0 | 0.0 | 0.0 | 0.0 | 0.0 | 0.0 | 0.0 | 0.0 | 0.0 | 0.0 | 0.0 | 0.0 | 0.07 | 0.0 | 0.0 | 0.0 | 0.0 | 0.0 | 0.0 | 0.0 | 0.0 | 0.0 | 0.0 | 0.0 | 0.0 | 0.0 | 0.0 | 0.0 | 0.0 | 0.0 |
| Leek | 0.0 | 0.0 | 0.0 | 0.0 | 0.0 | 0.0 | 0.0 | 0.0 | 0.0 | 0.0 | 0.0 | 0.0 | 0.0 | 0.9 | 0.0 | 0.0 | 0.0 | 0.0 | 0.0 | 0.0 | 0.0 | 0.0 | 0.0 | 0.0 | 0.0 | 0.0 | 0.0 | 0.1 | 0.0 | 0.0 | 0.0 | 0.0 | 0.0 | 0.0 | 0.0 | 0.0 | 0.0 | 0.0 | 0.0 | 0.0 | 0.0 | 0.0 | 0.0 |
| Lemon | 0.05 | 0.0 | 0.0 | 0.0 | 0.0 | 0.0 | 0.0 | 0.0 | 0.0 | 0.0 | 0.0 | 0.0 | 0.0 | 0.0 | 0.59 | 0.05 | 0.0 | 0.0 | 0.0 | 0.0 | 0.0 | 0.0 | 0.0 | 0.05 | 0.0 | 0.0 | 0.0 | 0.0 | 0.0 | 0.0 | 0.0 | 0.0 | 0.0 | 0.0 | 0.0 | 0.0 | 0.0 | 0.0 | 0.0 | 0.0 | 0.17 | 0.1 | 0.0 |
| Lime | 0.07 | 0.0 | 0.0 | 0.0 | 0.0 | 0.0 | 0.0 | 0.0 | 0.0 | 0.0 | 0.0 | 0.0 | 0.0 | 0.0 | 0.0 | 0.8 | 0.07 | 0.0 | 0.0 | 0.0 | 0.0 | 0.0 | 0.0 | 0.0 | 0.0 | 0.0 | 0.0 | 0.0 | 0.0 | 0.07 | 0.0 | 0.0 | 0.0 | 0.0 | 0.0 | 0.0 | 0.0 | 0.0 | 0.0 | 0.0 | 0.0 | 0.0 | 0.0 |
| Mango | 0.19 | 0.0 | 0.0 | 0.0 | 0.0 | 0.0 | 0.0 | 0.0 | 0.0 | 0.0 | 0.0 | 0.0 | 0.0 | 0.0 | 0.0 | 0.03 | 0.71 | 0.0 | 0.0 | 0.0 | 0.0 | 0.0 | 0.0 | 0.03 | 0.0 | 0.0 | 0.0 | 0.0 | 0.0 | 0.03 | 0.0 | 0.0 | 0.0 | 0.0 | 0.0 | 0.0 | 0.0 | 0.01 | 0.0 | 0.0 | 0.0 | 0.0 | 0.0 |
| Melon | 0.0 | 0.0 | 0.0 | 0.0 | 0.0 | 0.0 | 0.0 | 0.0 | 0.01 | 0.0 | 0.0 | 0.0 | 0.0 | 0.0 | 0.0 | 0.01 | 0.0 | 0.93 | 0.0 | 0.0 | 0.0 | 0.0 | 0.0 | 0.05 | 0.0 | 0.0 | 0.0 | 0.0 | 0.0 | 0.0 | 0.0 | 0.0 | 0.0 | 0.0 | 0.0 | 0.0 | 0.01 | 0.0 | 0.0 | 0.0 | 0.0 | 0.0 | 0.0 |
| Milk | 0.0 | 0.0 | 0.0 | 0.0 | 0.0 | 0.0 | 0.0 | 0.0 | 0.0 | 0.0 | 0.0 | 0.0 | 0.0 | 0.0 | 0.0 | 0.0 | 0.0 | 0.0 | 1.0 | 0.0 | 0.0 | 0.0 | 0.0 | 0.0 | 0.0 | 0.0 | 0.0 | 0.0 | 0.0 | 0.0 | 0.0 | 0.0 | 0.0 | 0.0 | 0.0 | 0.0 | 0.0 | 0.0 | 0.0 | 0.0 | 0.0 | 0.0 | 0.0 |
| Nectarine | 0.0 | 0.0 | 0.0 | 0.0 | 0.0 | 0.0 | 0.0 | 0.0 | 0.0 | 0.0 | 0.0 | 0.0 | 0.0 | 0.0 | 0.0 | 0.0 | 0.0 | 0.0 | 0.0 | 1.0 | 0.0 | 0.0 | 0.0 | 0.0 | 0.0 | 0.0 | 0.0 | 0.0 | 0.0 | 0.0 | 0.0 | 0.0 | 0.0 | 0.0 | 0.0 | 0.0 | 0.0 | 0.0 | 0.0 | 0.0 | 0.0 | 0.0 | 0.0 |
| Oat-Milk | 0.0 | 0.0 | 0.0 | 0.0 | 0.0 | 0.0 | 0.0 | 0.0 | 0.0 | 0.0 | 0.0 | 0.0 | 0.0 | 0.0 | 0.0 | 0.0 | 0.0 | 0.0 | 0.0 | 0.0 | 1.0 | 0.0 | 0.0 | 0.0 | 0.0 | 0.0 | 0.0 | 0.0 | 0.0 | 0.0 | 0.0 | 0.0 | 0.0 | 0.0 | 0.0 | 0.0 | 0.0 | 0.0 | 0.0 | 0.0 | 0.0 | 0.0 | 0.0 |
| Oatghurt | 0.0 | 0.0 | 0.0 | 0.0 | 0.0 | 0.0 | 0.0 | 0.0 | 0.0 | 0.0 | 0.0 | 0.0 | 0.0 | 0.0 | 0.0 | 0.0 | 0.0 | 0.0 | 0.0 | 0.0 | 0.0 | 1.0 | 0.0 | 0.0 | 0.0 | 0.0 | 0.0 | 0.0 | 0.0 | 0.0 | 0.0 | 0.0 | 0.0 | 0.0 | 0.0 | 0.0 | 0.0 | 0.0 | 0.0 | 0.0 | 0.0 | 0.0 | 0.0 |
| Onion | 0.0 | 0.0 | 0.0 | 0.0 | 0.0 | 0.0 | 0.0 | 0.0 | 0.0 | 0.0 | 0.0 | 0.0 | 0.0 | 0.0 | 0.0 | 0.0 | 0.0 | 0.0 | 0.0 | 0.0 | 0.0 | 0.0 | 1.0 | 0.0 | 0.0 | 0.0 | 0.0 | 0.0 | 0.0 | 0.0 | 0.0 | 0.0 | 0.0 | 0.0 | 0.0 | 0.0 | 0.0 | 0.0 | 0.0 | 0.0 | 0.0 | 0.0 | 0.0 |
| Orange | 0.0 | 0.0 | 0.0 | 0.0 | 0.0 | 0.0 | 0.0 | 0.0 | 0.0 | 0.0 | 0.0 | 0.0 | 0.0 | 0.0 | 0.0 | 0.0 | 0.0 | 0.0 | 0.0 | 0.0 | 0.0 | 0.0 | 0.0 | 0.66 | 0.0 | 0.0 | 0.0 | 0.0 | 0.0 | 0.0 | 0.0 | 0.0 | 0.0 | 0.09 | 0.25 | 0.0 | 0.0 | 0.0 | 0.0 | 0.0 | 0.0 | 0.0 | 0.0 |
| Papaya | 0.0 | 0.0 | 0.0 | 0.0 | 0.0 | 0.0 | 0.0 | 0.0 | 0.0 | 0.0 | 0.0 | 0.0 | 0.0 | 0.0 | 0.0 | 0.0 | 0.0 | 0.0 | 0.0 | 0.0 | 0.0 | 0.0 | 0.0 | 0.0 | 1.0 | 0.0 | 0.0 | 0.0 | 0.0 | 0.0 | 0.0 | 0.0 | 0.0 | 0.0 | 0.0 | 0.0 | 0.0 | 0.0 | 0.0 | 0.0 | 0.0 | 0.0 | 0.0 |
| Passion-Fruit | 0.0 | 0.0 | 0.22 | 0.0 | 0.0 | 0.0 | 0.0 | 0.0 | 0.0 | 0.0 | 0.0 | 0.0 | 0.0 | 0.0 | 0.0 | 0.0 | 0.04 | 0.0 | 0.0 | 0.0 | 0.0 | 0.0 | 0.0 | 0.0 | 0.0 | 0.63 | 0.0 | 0.0 | 0.0 | 0.0 | 0.0 | 0.0 | 0.0 | 0.0 | 0.0 | 0.11 | 0.0 | 0.0 | 0.0 | 0.0 | 0.0 | 0.0 | 0.0 |
| Peach | 0.0 | 0.0 | 0.0 | 0.0 | 0.0 | 0.0 | 0.0 | 0.0 | 0.0 | 0.0 | 0.0 | 0.0 | 0.0 | 0.0 | 0.0 | 0.0 | 0.0 | 0.0 | 0.0 | 0.0 | 0.0 | 0.0 | 0.0 | 0.0 | 0.0 | 0.0 | 1.0 | 0.0 | 0.0 | 0.0 | 0.0 | 0.0 | 0.0 | 0.0 | 0.0 | 0.0 | 0.0 | 0.0 | 0.0 | 0.0 | 0.0 | 0.0 | 0.0 |
| Pear | 0.04 | 0.0 | 0.0 | 0.0 | 0.01 | 0.0 | 0.0 | 0.0 | 0.0 | 0.0 | 0.0 | 0.0 | 0.0 | 0.0 | 0.0 | 0.0 | 0.01 | 0.0 | 0.0 | 0.0 | 0.0 | 0.0 | 0.0 | 0.01 | 0.0 | 0.0 | 0.0 | 0.94 | 0.0 | 0.0 | 0.0 | 0.0 | 0.0 | 0.0 | 0.0 | 0.0 | 0.0 | 0.0 | 0.0 | 0.0 | 0.01 | 0.0 | 0.0 |
| Pepper | 0.0 | 0.0 | 0.0 | 0.0 | 0.0 | 0.0 | 0.0 | 0.0 | 0.0 | 0.0 | 0.0 | 0.0 | 0.0 | 0.0 | 0.0 | 0.0 | 0.0 | 0.0 | 0.0 | 0.0 | 0.0 | 0.0 | 0.0 | 0.0 | 0.0 | 0.0 | 0.0 | 0.0 | 0.99 | 0.0 | 0.0 | 0.0 | 0.0 | 0.0 | 0.0 | 0.0 | 0.0 | 0.0 | 0.0 | 0.0 | 0.01 | 0.0 | 0.0 |
| Pineapple | 0.0 | 0.0 | 0.0 | 0.0 | 0.0 | 0.0 | 0.0 | 0.0 | 0.0 | 0.0 | 0.0 | 0.0 | 0.04 | 0.0 | 0.0 | 0.0 | 0.0 | 0.0 | 0.0 | 0.0 | 0.0 | 0.0 | 0.0 | 0.0 | 0.0 | 0.0 | 0.0 | 0.0 | 0.0 | 0.96 | 0.0 | 0.0 | 0.0 | 0.0 | 0.0 | 0.0 | 0.0 | 0.0 | 0.0 | 0.0 | 0.0 | 0.0 | 0.0 |
| Plum | 0.0 | 0.0 | 0.0 | 0.0 | 0.0 | 0.0 | 0.0 | 0.0 | 0.0 | 0.0 | 0.0 | 0.0 | 0.0 | 0.0 | 0.0 | 0.0 | 0.0 | 0.0 | 0.0 | 0.0 | 0.0 | 0.0 | 0.0 | 0.0 | 0.0 | 0.0 | 0.0 | 0.0 | 0.0 | 0.0 | 1.0 | 0.0 | 0.0 | 0.0 | 0.0 | 0.0 | 0.0 | 0.0 | 0.0 | 0.0 | 0.0 | 0.0 | 0.0 |
| Pomegranate | 0.0 | 0.0 | 0.0 | 0.0 | 0.0 | 0.0 | 0.0 | 0.0 | 0.0 | 0.0 | 0.0 | 0.0 | 0.0 | 0.0 | 0.0 | 0.0 | 0.0 | 0.0 | 0.0 | 0.0 | 0.0 | 0.0 | 0.0 | 0.04 | 0.0 | 0.0 | 0.0 | 0.0 | 0.0 | 0.0 | 0.0 | 0.92 | 0.04 | 0.0 | 0.0 | 0.0 | 0.0 | 0.0 | 0.0 | 0.0 | 0.0 | 0.0 | 0.0 |
| Potato | 0.0 | 0.0 | 0.0 | 0.0 | 0.0 | 0.0 | 0.0 | 0.0 | 0.0 | 0.0 | 0.0 | 0.0 | 0.0 | 0.0 | 0.0 | 0.0 | 0.0 | 0.0 | 0.0 | 0.0 | 0.0 | 0.0 | 0.0 | 0.0 | 0.0 | 0.0 | 0.0 | 0.0 | 0.0 | 0.0 | 0.0 | 0.0 | 1.0 | 0.0 | 0.0 | 0.0 | 0.0 | 0.0 | 0.0 | 0.0 | 0.0 | 0.0 | 0.0 |
| Red-Beet | 0.0 | 0.0 | 0.0 | 0.0 | 0.0 | 0.0 | 0.0 | 0.0 | 0.0 | 0.0 | 0.0 | 0.0 | 0.0 | 0.0 | 0.0 | 0.0 | 0.0 | 0.0 | 0.0 | 0.0 | 0.0 | 0.0 | 0.0 | 0.0 | 0.0 | 0.0 | 0.0 | 0.0 | 0.0 | 0.0 | 0.0 | 0.0 | 0.0 | 1.0 | 0.0 | 0.0 | 0.0 | 0.0 | 0.0 | 0.0 | 0.0 | 0.0 | 0.0 |
| Red-Grapefruit | 0.0 | 0.0 | 0.0 | 0.0 | 0.0 | 0.0 | 0.0 | 0.0 | 0.0 | 0.0 | 0.0 | 0.0 | 0.0 | 0.0 | 0.0 | 0.0 | 0.03 | 0.0 | 0.0 | 0.0 | 0.0 | 0.0 | 0.0 | 0.0 | 0.0 | 0.15 | 0.0 | 0.0 | 0.0 | 0.0 | 0.0 | 0.0 | 0.0 | 0.0 | 0.62 | 0.21 | 0.0 | 0.0 | 0.0 | 0.0 | 0.0 | 0.0 | 0.0 |
| Satsumas | 0.0 | 0.0 | 0.0 | 0.0 | 0.0 | 0.0 | 0.0 | 0.0 | 0.0 | 0.0 | 0.0 | 0.0 | 0.0 | 0.0 | 0.0 | 0.0 | 0.0 | 0.0 | 0.0 | 0.0 | 0.0 | 0.0 | 0.0 | 0.16 | 0.0 | 0.0 | 0.0 | 0.0 | 0.0 | 0.0 | 0.0 | 0.0 | 0.0 | 0.0 | 0.0 | 0.84 | 0.0 | 0.0 | 0.0 | 0.0 | 0.0 | 0.0 | 0.0 |
| Sour-Cream | 0.0 | 0.0 | 0.0 | 0.0 | 0.0 | 0.0 | 0.0 | 0.0 | 0.0 | 0.0 | 0.0 | 0.0 | 0.0 | 0.0 | 0.0 | 0.0 | 0.0 | 0.0 | 0.0 | 0.0 | 0.0 | 0.02 | 0.0 | 0.0 | 0.0 | 0.0 | 0.0 | 0.0 | 0.0 | 0.0 | 0.0 | 0.0 | 0.0 | 0.0 | 0.0 | 0.0 | 0.93 | 0.05 | 0.0 | 0.0 | 0.0 | 0.0 | 0.0 |
| Sour-Milk | 0.0 | 0.0 | 0.0 | 0.0 | 0.0 | 0.0 | 0.0 | 0.0 | 0.0 | 0.0 | 0.0 | 0.0 | 0.0 | 0.0 | 0.0 | 0.0 | 0.0 | 0.0 | 0.0 | 0.0 | 0.0 | 0.0 | 0.0 | 0.0 | 0.0 | 0.0 | 0.0 | 0.0 | 0.0 | 0.0 | 0.0 | 0.0 | 0.0 | 0.0 | 0.0 | 0.0 | 0.0 | 1.0 | 0.0 | 0.0 | 0.0 | 0.0 | 0.0 |
| Soy-Milk | 0.0 | 0.0 | 0.0 | 0.0 | 0.0 | 0.0 | 0.0 | 0.0 | 0.0 | 0.0 | 0.0 | 0.0 | 0.0 | 0.0 | 0.0 | 0.0 | 0.0 | 0.0 | 0.0 | 0.0 | 0.0 | 0.0 | 0.0 | 0.0 | 0.0 | 0.0 | 0.0 | 0.0 | 0.0 | 0.0 | 0.0 | 0.0 | 0.0 | 0.0 | 0.0 | 0.0 | 0.0 | 0.0 | 1.0 | 0.0 | 0.0 | 0.0 | 0.0 |
| Soyghurt | 0.0 | 0.0 | 0.0 | 0.0 | 0.0 | 0.0 | 0.0 | 0.0 | 0.0 | 0.0 | 0.0 | 0.0 | 0.0 | 0.0 | 0.0 | 0.0 | 0.0 | 0.0 | 0.0 | 0.0 | 0.0 | 0.0 | 0.0 | 0.0 | 0.0 | 0.0 | 0.0 | 0.0 | 0.0 | 0.0 | 0.0 | 0.0 | 0.0 | 0.0 | 0.0 | 0.0 | 0.0 | 0.0 | 0.0 | 1.0 | 0.0 | 0.0 | 0.0 |
| Tomato | 0.0 | 0.0 | 0.0 | 0.0 | 0.0 | 0.0 | 0.0 | 0.0 | 0.0 | 0.0 | 0.0 | 0.0 | 0.0 | 0.0 | 0.0 | 0.0 | 0.0 | 0.0 | 0.0 | 0.0 | 0.0 | 0.0 | 0.0 | 0.0 | 0.0 | 0.0 | 0.0 | 0.0 | 0.0 | 0.0 | 0.0 | 0.0 | 0.0 | 0.0 | 0.0 | 0.0 | 0.0 | 0.0 | 0.0 | 0.0 | 1.0 | 0.0 | 0.0 |
| Yoghurt | 0.0 | 0.0 | 0.0 | 0.0 | 0.0 | 0.0 | 0.0 | 0.0 | 0.0 | 0.0 | 0.0 | 0.0 | 0.0 | 0.0 | 0.0 | 0.0 | 0.0 | 0.0 | 0.0 | 0.0 | 0.0 | 0.0 | 0.0 | 0.0 | 0.0 | 0.0 | 0.0 | 0.0 | 0.0 | 0.0 | 0.0 | 0.0 | 0.0 | 0.0 | 0.0 | 0.0 | 0.0 | 0.0 | 0.0 | 0.0 | 0.0 | 1.0 | 0.0 |
| Zucchini | 0.0 | 0.0 | 0.0 | 0.0 | 0.0 | 0.0 | 0.0 | 0.0 | 0.0 | 0.0 | 0.0 | 0.0 | 0.0 | 0.0 | 0.0 | 0.0 | 0.0 | 0.0 | 0.0 | 0.0 | 0.0 | 0.0 | 0.0 | 0.0 | 0.0 | 0.0 | 0.0 | 0.0 | 0.0 | 0.0 | 0.0 | 0.0 | 0.0 | 0.0 | 0.0 | 0.0 | 0.0 | 0.0 | 0.0 | 0.0 | 0.0 | 0.0 | 1.0 |

## 4.2. Integration of Multiple Classifiers into a Siamese Network

Three approaches based on Siamese nets have been evaluated: a traditional Siamese net with a CNN, a Siamese net using a LOMO descriptor [2] and a Siamese net integrating a CNN and a LOMO descriptor.

In order to train these models, training and test data from the original dataset have initially been placed together. This is because the original division of the dataset is prepared for classification. However, for the OSL problem, the list of classes has been completely split into training, validation and test, as explained in the analysis. Then, some classes have been moved to training (50), some to validation (18) and some to test (13). This division has been the same for the evaluation of the three models. During training, groups of pairs of images are processed in batches. For each image in the dataset, a positive pair and a negative pair have been randomly chosen. In the positive pair, there is an image from a class and a different random image from the same class. In the negative pair, there is an image from a class and another one from a different random category.

Figure 12a shows the training and validation accuracy for the Siamese net based on ResNeXt-101. Figure 12b shows the loss of this model. This model has been trained with an image shape of 299 × 299 for 30 epochs, and the weights for the first layers have been blocked using pre-trained weights for ImageNet [31]. The size of the batch has been 16, using a binary cross entropy loss and an Adam optimizer with a learning rate of 0.001. The best validation accuracy has been obtained in epoch five. The same parameters have been used for the LOMO and the ResNeXt-101+LOMO models.

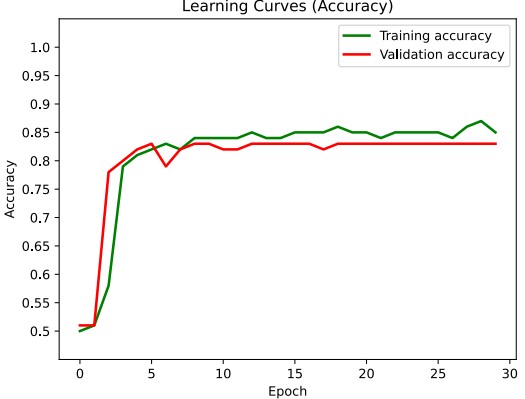

(**a**). Training/validation accuracy for ResNeXt-101-based Siamese network.

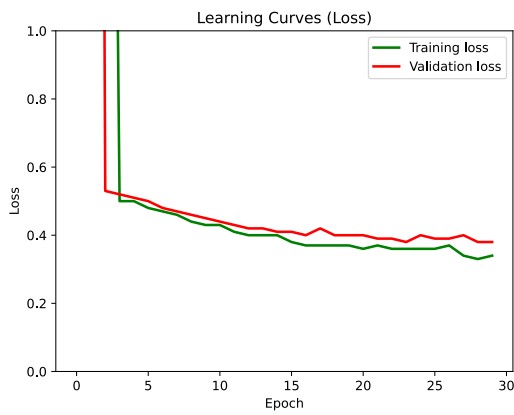

(**b**). Training/validation loss for ResNeXt-101-based Siamese network.

**Figure 12.** Siamese network training with ResNeXt-101.

Figure 13a shows the training and validation accuracy for the Siamese net based on LOMO. Figure 13b shows the loss of this model. The descriptors for each image are pre-calculated since their parameters are not trainable within the model. The best validation accuracy has been obtained in epoch 29.

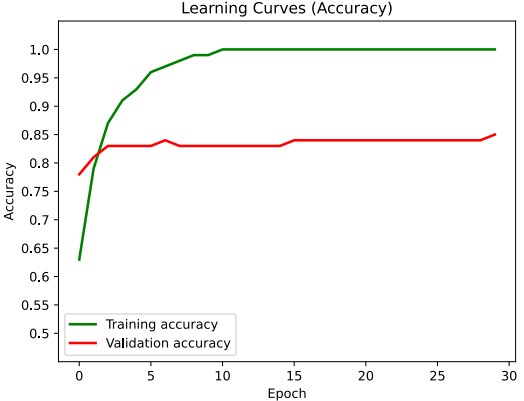

(**a**). Training/validation accuracy for LOMO-based Siamese network.

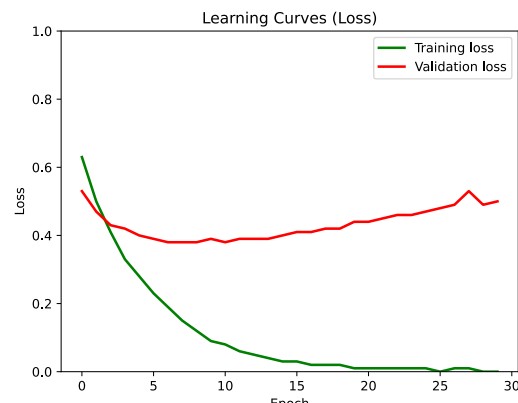

(**b**). Training/validation loss for LOMO-based Siamese network.

**Figure 13.** Siamese network training with LOMO.

Figure 14a shows the training and validation accuracy for the Siamese net based on ResNeXt-101+LOMO. Figure 14b shows the loss of this model. The best validation accuracy has been obtained in epoch five. Again, the LOMO descriptors for each image are pre-calculated and the ResNeXt-101 part received images with shape $299 \times 299$.

These last graphs allow us to observe how the integration of the two sub-models within the Siamese model improves the behavior of the two previous models. Thus, for example, the validation accuracy increases with respect to the two previous models while the loss stabilizes, which is something that did not occur in the LOMO model. The validation loss increased from epoch eight onward in that model.

Another modification has included an L2-regularization in the *dense* layer. This technique is used to reduce model overfitting. The regularization has applied a penalty on the layer's kernel and on the layer's bias. For both penalties, the L2-factor has been 0.001. This regularization has reduced the gap between training and validation. Figure 15a shows the training and validation accuracy for the Siamese net based on ResNeXt-101+LOMO

with L2-regularization. Figure 15b shows the loss of this model. The best validation accuracy has been obtained in epoch 10, with a maximum accuracy value of 0.95 and a maximum validation accuracy of 0.89.

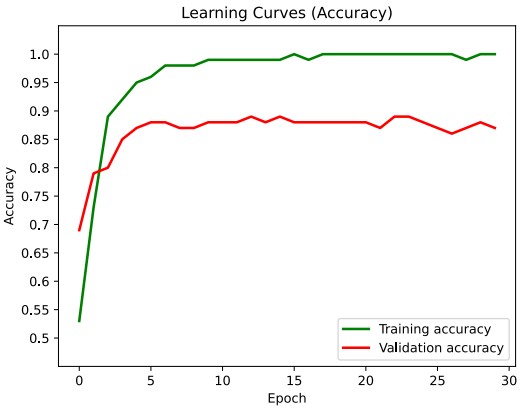

(**a**). Training/validation accuracy for ResNeXt-101+LOMO-based Siamese network.

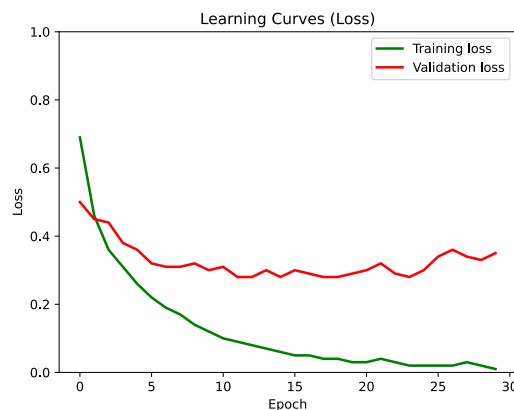

(**b**). Training/validation loss for ResNeXt-101+LOMO-based Siamese network.

**Figure 14.** Siamese network training with ResNeXt-101+LOMO.

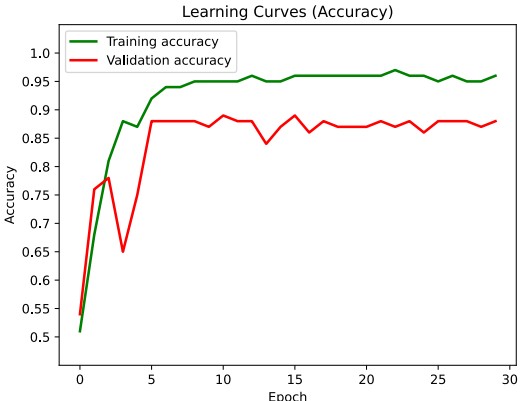

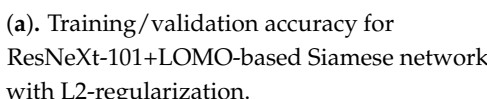

(**a**). Training/validation accuracy for ResNeXt-101+LOMO-based Siamese network with L2-regularization.

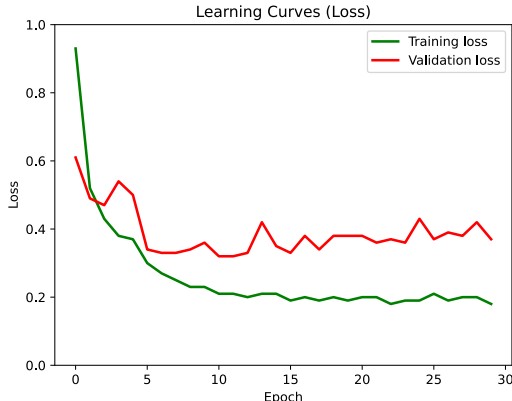

(**b**). Training/validation loss for ResNeXt-101+LOMO-based Siamese network with L2-regularization.

**Figure 15.** Siamese network training with ResNeXt-101+LOMO and L2-regularization.

The results of the four models are shown in Table 4. The ResNeXt-101+LOMO composite model improves the results of the other two models. The evaluation has been carried out by means of five random test groups (*k* = 5) where there are positive and negative pairs of images. The neural network learns to consider the best aspects of each sub-model to globally improve the result. Since the test set is previously balanced, the result of the test accuracy and balanced test accuracy is equal.

The best results are produced when using regularization in the model. Table 5 shows the success rate for each class with pairs of images from the same and different category. Although the negative detection performance of some classes is worse in the ResNeXt-101+LOMO model than in the ResNeXt-101 alone, the overall values improve, as previously shown. These combined models significantly improve recall, drastically reducing the false negatives that occur when comparing images of the same category. In the case of using L2-regularization, the recall improves even more but at the cost of penalizing the precision as the number of false positives increases. For this reason, the negative pair detection

performs slightly worse as the detection of false positives increases. However, the precision, the balanced accuracy and F1-score are globally improved.

**Table 4.** Comparison of different proposed Siamese networks.

| Metric | ResNeXt-101 | LOMO | ResNeXt-101 + LOMO | ResNeXt-101 + LOMO (with L2-Regularization) |
|---|---|---|---|---|
| **Test accuracy** *(k = 5)* | 84.5 | 78.0 | 86.5 | 88.2 |
| **Balanced Test accuracy** *(k = 5)* | 84.5 | 78.0 | 86.5 | 88.2 |
| **Precision** *(k = 5)* | 83.9 | 73.6 | 84.6 | 82.7 |
| **Recall** *(k = 5)* | 85.5 | 87.3 | 89.2 | 96.5 |
| **F1-score** *(k = 5)* | 84.7 | 79.9 | 86.8 | 89.1 |

**Table 5.** Success rate by category of the proposed Siamese networks (%).

| Test Category | ResNeXt-101 Positive Pairs | ResNeXt-101 Negative Pairs | LOMO Positive Pairs | LOMO Negative Pairs | ResNeXt-101 + LOMO Positive Pairs | ResNeXt-101 + LOMO Negative Pairs | ResNeXt-101 + LOMO Positive Pairs (with L2-Regularization) | ResNeXt-101 + LOMO Negative Pairs (with L2-Regularization) |
|---|---|---|---|---|---|---|---|---|
| Arla-Ecological-Sour-Cream | 97.31 | 83.85 | 94.23 | 84.62 | 94.23 | 83.08 | 99.62 | 85.77 |
| Brown-Cap-Mushroom | 64.87 | 88.46 | 61.28 | 63.33 | 87.18 | 87.95 | 93.59 | 80.26 |
| Carrots | 72.24 | 99.29 | 88.0 | 61.65 | 92.0 | 95.76 | 92.94 | 82.12 |
| God-Morgon-Orange-Juice | 88.0 | 82.67 | 96.44 | 60.0 | 95.11 | 93.33 | 99.56 | 83.56 |
| God-Morgon-Red-Grapefruit-Juice | 97.0 | 79.0 | 100.0 | 61.5 | 98.5 | 87.5 | 100.0 | 73.5 |
| Granny-Smith | 97.78 | 84.44 | 84.27 | 78.97 | 85.81 | 83.25 | 95.9 | 86.84 |
| Lime | 94.1 | 77.38 | 76.39 | 80.0 | 85.57 | 82.3 | 97.38 | 69.84 |
| Mango | 99.05 | 71.11 | 75.24 | 66.03 | 82.22 | 73.33 | 99.05 | 70.16 |
| Oatly-Oat-Milk | 86.67 | 85.08 | 95.87 | 82.86 | 86.03 | 84.44 | 98.41 | 83.81 |
| Passion-Fruit | 46.91 | 85.09 | 89.09 | 51.64 | 83.27 | 76.36 | 88.0 | 78.91 |
| Sweet-Potato | 86.43 | 75.71 | 98.21 | 54.29 | 91.43 | 74.64 | 97.86 | 72.5 |
| Tropicana-Golden-Grapefruit | 99.55 | 82.73 | 99.55 | 74.55 | 91.36 | 87.27 | 99.09 | 87.27 |
| Yellow-Onion | 86.67 | 82.13 | 96.8 | 64.27 | 94.67 | 78.67 | 98.4 | 76.8 |

The training sessions have been carried out on an i9-10900K server with 128 GB RAM and 2 GPU RTX-3090 with 24 GB GDDR6X. This server required 187 min for training the ResNeXt-101+LOMO-based Siamese net.

### 4.3. Results Discussion

Two different sorts of experiments have been carried out. The first experiment improved the baseline classification set by the authors of the Grocery Store Dataset [3]. They obtained a test accuracy of 85% using a DenseNet-169 model. A ResNeXt-101 with cardinality 32, a bottleneck width of eight [4] and a 20% dropout layer has improved the result up to 91.60% test accuracy. As the model was confused between oranges and satsumas, another tiered model was trained and added to reduce this error. By staggering the two models, the test accuracy has increased to 92%. This cascade modeling technique allows improving the model performance when there is a certain error in the detection of some specific classes.

Recently, a paper published by Leo et al. [40] has surpassed these values by using a technique known as *Ensemble of networks*. Considered an ensemble of *M* models, this technique chooses the output according to a maximum number of occurrences among the *M* individual networks. As an example, the *Ensemble C* uses the following models simultaneously: ResNet-50, ResNet-101, ResNet-152, EfficientNet-b1, DenseNet-121, DenseNet-161 and DenseNet-201. The problem with this approach is that it requires the training of a large group of models, not only increasing the time in training but also in prediction.

The second experiment has consisted in integrating multiple classifiers into a Siamese network in order to show how this new method improves the current OSL techniques. The results of four Siamese models have been shown, one including a ResNeXt-101 in each sister, one using a LOMO descriptor, one including a ResNeXt-101 together with a LOMO descriptor and one similar to the previous model but using a L2-regularization.

The L2-regularization has been used to reduce the overfitting detected in the model with ResNeXt-101 and LOMO.

The integration of a descriptor and a CNN into the Siamese improves the model initially defined by Koch et al. [18] and can be easily applied to other problems, such as medical image classification [41] or people re-identification [42]. The F1-score obtained using a traditional Siamese with a deep CNN has been 84.5%, while the score obtained with the composite architecture has been 89.1%. This technique is open to using different descriptors or classifiers and even can integrate more than two within the same network.

## 5. Conclusions

This paper presents the integration of a classifier and a descriptor generator into a Siamese network. This promising combination of techniques within a Siamese net allows solving the problem of one-shot learning. The OSL problem uses a single image per category to classify new images in a dataset. The chosen dataset is very interesting for this problem because grocery products can be classified with a single image. This problem has been traditionally solved by means of data augmentation, including Generative Adversarial Networks or feature descriptors and matching. Siamese nets solve the problem in a different way, emulating the behavior of humans. People are able to distinguish another person they have only seen once, but they are not able to distinguish a monkey. This is because our brains have been trained by the observation of many people throughout our lives.

Siamese nets are usually composed of two sisters that implement convolutional networks, sharing the same weights. In this article, a LOMO descriptor, together with a ResNeXt-101, has been integrated within the Siamese, showing how the results improve with respect to the use of a single classifier. The LOMO descriptor was previously created for people re-identification and is robust against color and viewpoint changes. The model is trained with a group of categories, validated with another group of categories and finally evaluated with another different group. The data are split into positive and negative pairs, where the positive pair represents two images that belong to the same category and the negative pair to a different one. The evaluation has been carried out by means of five random test groups where there are positive and negative pairs of images. In order to carry out the choice of the convolutional part of the model, the baseline classification of the used dataset has been improved. The computational cost of this technique is as low as descriptor generation, and image inference can be performed in real time.

The idea of using a descriptor together with a CNN is open to using multiple classifiers/descriptors simultaneously within the Siamese. Other authors have combined multiple classifiers for a classification problem, and it shows promising results. Future works will consist in integrating multiple classifiers and different descriptors into a Siamese model and to try to solve different types of problems where these techniques can be used, such as the case of re-identification of people.

**Author Contributions:** J.D.D. contributed to the entire work, designed the experiments, analyzed the results and prepared the paper. R.M.A. contributed to the work by analyzing the results and preparing the paper. L.M.G.R. contributed as the scientific director, monitoring the work progress and preparing the paper. All authors have read and agreed to the published version of the manuscript.

**Funding:** The present research has been partially funded by "Instituto para la Competitividad Empresarial de Castilla y León" (Project CCTT3/20/VA/0003) through the line "2020 Proyectos I+D orientados a la excelencia y mejora competitiva de los CCTT", cofinancied with FEDER funds, and by "Centro para el Desarrollo Tecnológico Industrial (CDTI)" (Ref. CER-20211007) through the line "Convocatoria del programa Cervera para Centros Tecnológicos 2020".

**Conflicts of Interest:** The authors declare no conflict of interest regarding the publication of this manuscript.

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
