# Peer review of "Improvement of One-Shot-Learning by Integrating a Convolutional Neural Network and an Image Descriptor into a Siamese Neural Network"

_applsci, doi:10.3390/app11177839_

Round 1

Reviewer 1 Report

The topic is interesting. However, the presentation requires considerable revision. Therefore, I have the following suggestions for improving the quality of the paper.

  1. There are many grammatical mistakes, few I have mentioned in the review report. (attached as pdf).
  2. The equation used in the paper should be explained in the text for the reader to understand easily and reuse them.
  3. The discussion part is missing and mixed up with the experiment and conclusion sections. However, in a good research paper, there is always a discussion part.
  4. Rearrange the sections in the paper to make it comprehensive and will be a proper flow of information.

Thank you and best luck.

Author Response

Please see attach PDF file.

Reviewer 2 Report

This paper proposes a method that integrates a CNN and a LOMO descriptor into a Siamese neural network for one-shot grocery image classification.
I think the aim of this paper is not to propose a novel method but a practical one. In this regard, a promising network architecture for one-shot grocery image classification was experimentally investigated. 
The experiments and evaluations were not fully performed, but sufficient to understand that the proposed architecture was effective.

I suggest that the authors consider the followings for revision:
In Section 2, related works to different topics were described together. So, I think it would be better to divide those works into subsections.
In page 7, what do the super- and sub-scripts of the two parameters for representing the SILTP histogram scales mean? To be more self-contained, I think it would be better to detail how to compute the SILTP descriptor, if possible, visually.
In page 7, why are the numbers of neurons (2000, 1000, 500, ..... 4, 2)?
Figures 11 through 14 do not have captions.
I think it would be better to leave out Figure 15.
There is no discussion for the results in Table 5. Please add it.
In Table 5, the accuracy of "ResNet + LOMO" for negative pairs decreased with L2-regularization. Please explain why.

Author Response

Please see attach PDF file.

Round 2

Reviewer 1 Report

Dear authors,

My aspects and comments of the previous review were appropriately adapted or rejected for good reasons. Figures were upgraded and content reasonable extended. The discussion part is included.

Good Luck & Best Wishes.